# TOWARD COMPLEX-VALUED NEURAL NETWORKS FOR WAVEFORM GENERATION

**Hyung-Seok Oh, Deok-Hyeon Cho, Seung-Bin Kim & Seong-Whan Lee** [*]
Department of Artificial Intelligence
Korea University
Seoul, Republic of Korea
`{hs_oh, dh_cho, sb-kim, sw.lee}@korea.ac.kr`

## ABSTRACT

Neural vocoders have recently advanced waveform generation, yielding natural and expressive audio. Among these approaches, iSTFT-based vocoders have recently gained attention. They predict a complex-valued spectrogram and then synthesize the waveform via iSTFT, thereby avoiding learned upsampling stages that can increase computational cost. However, current approaches use real-valued networks that process the real and imaginary parts independently. This separation limits their ability to capture the inherent structure of complex spectrograms. We present ComVo, a **Com**plex-valued neural **Vo**coder whose generator and discriminator use native complex arithmetic. This enables an adversarial training framework that provides structured feedback in complex-valued representations. To guide phase transformations in a structured manner, we introduce phase quantization, which discretizes phase values and regularizes the training process. Finally, we propose a block-matrix computation scheme to improve training efficiency by reducing redundant operations. Experiments demonstrate that ComVo achieves higher synthesis quality than comparable real-valued baselines, and that its block-matrix scheme reduces training time by 25%. Audio samples and code are available at `https://hs-oh-prml.github.io/ComVo/`.

## 1 INTRODUCTION

Deep learning-based vocoders have significantly advanced speech synthesis, producing more natural and expressive synthetic speech. Recent developments include models based on generative adversarial networks (GANs) (Kumar et al., 2019; Yamamoto et al., 2020; Kong et al., 2020; Lee et al., 2023), normalizing flow-based models (van den Oord et al., 2018; Ping et al., 2020; Lee et al., 2020), and diffusion-based models (Kong et al., 2021; Lee et al., 2022; Chen et al., 2021; Lee et al., 2025). Although these approaches achieve high-fidelity speech generation, some neural vocoders still rely on sequential sample prediction or learned upsampling, thereby increasing model complexity and inference latency.

An alternative is to synthesize speech in the spectral domain using the inverse short-time Fourier transform (iSTFT). Operating directly on complex spectrograms (Oyamada et al., 2018; Neekhara et al., 2019; Gritsenko et al., 2020; Kaneko et al., 2022; 2023; Siuzdak, 2024; Yoneyama et al., 2024; Liu et al., 2025) avoids the need for sample-by-sample generation and learned upsampling. To our knowledge, current iSTFT-based vocoders rely on real-valued neural networks (RVNNs) that process real and imaginary parts as separate channels. This separation limits their ability to model the coupling between these components.

Complex-valued neural networks (CVNNs) extend standard neural networks to the complex domain by allowing both inputs and parameters to be complex-valued. Operating entirely in the complex domain enables these models to capture the intrinsic dependencies between the real and imaginary components. CVNNs have been applied in domains such as radar signal classification (Yang et al., 2022), MRI reconstruction (Vasudeva et al., 2022), and wireless communication (Xu et al., 2022),

---

[*]Corresponding author

where measurements carry both magnitude and phase information and naturally form complex-valued data. In speech processing, CVNNs have been explored for tasks including speech enhancement (Nustede & Anemüller, 2024; Mamun & Hansen, 2023), speech recognition (Hayakawa et al., 2018), and even statistical parametric speech synthesis (Hu et al., 2016). These studies demonstrate the potential of CVNNs to better capture spectral structure.

Although some recent vocoders produce complex spectrograms, they still use real-valued networks that handle each spectrogram channel independently. CVNNs, by jointly processing complex coefficients, could overcome this limitation. By treating each spectrogram coefficient as a unified complex entity, CVNN-based models can capture cross-component interactions that real-valued models miss. Motivated by this, we adopt CVNNs to better capture structure in the complex domain, yielding higher-quality synthesis.

In this work, we propose ComVo, a **Com**plex-valued neural **Vo**coder that performs iSTFT-based waveform generation entirely in the complex domain with a GAN-based architecture. The generator uses CVNN layers to jointly model the real and imaginary components of spectrograms, thereby better capturing their algebraic structure. We then design a complex multi-resolution discriminator (cMRD) that operates directly on complex spectrograms. Together, these components form a complex-domain adversarial training framework in which both the generator and discriminator operate on complex-valued representations. This design allows feedback that respects the structure of the complex domain. Inspired by recent studies on complex activation functions (Vasudeva et al., 2022), we introduce phase quantization, a nonlinear transformation that discretizes phase angles to serve as an inductive bias for stable learning. Finally, to reduce redundant computations in complex-valued operations, we develop a block-matrix computation scheme that improves overall training efficiency.

- **CVNN-based architecture with complex adversarial training:** We introduce ComVo, which, to our knowledge, is the first iSTFT-based vocoder to employ complex-valued neural networks in both its generator and discriminator. We design the discriminator losses in the complex domain, thus establishing an adversarial framework that operates on complex-valued representations.

- **Structured nonlinear transformation:** We propose phase quantization, a tailored nonlinear operation that discretizes phase angles and serves as an inductive bias.

- **Block-matrix computation scheme:** We present an efficient implementation that fuses the four real-valued multiplications required for each complex operation into a single block-matrix multiplication, reducing training time by 25%.

- **Improved synthesis performance:** ComVo outperforms real-valued vocoders, as demonstrated in our experiments.

## 2 RELATED WORKS

### 2.1 COMPLEX-VALUED NEURAL NETWORKS

CVNNs represent inputs, activations, and weights directly as complex numbers. They have been applied in a range of domains where signals are naturally expressed in the complex field, including radar classification (Yang et al., 2022), MRI reconstruction (Vasudeva et al., 2022), wireless communication (Xu et al., 2022), and audio analysis (Sarroff, 2018). Several studies report that CVNNs can exhibit favorable learning behavior or approximation properties compared to real-valued networks in various settings (Barrachina et al., 2021; Voigtlaender, 2023; Geuchen & Voigtlaender, 2023). This prior work suggests that complex-valued modeling can be a viable choice when dealing with data or transformations formulated in the complex domain.

### 2.2 ISTFT-BASED VOCODER

The short-time Fourier transform (STFT) decomposes a waveform into overlapping frames of complex spectral coefficients. The iSTFT reconstructs the time-domain signal using the overlap-add method. This fully differentiable analysis-synthesis pipeline enables end-to-end training on frame-level spectra while generating sample-level waveforms in a single pass. This approach eliminates

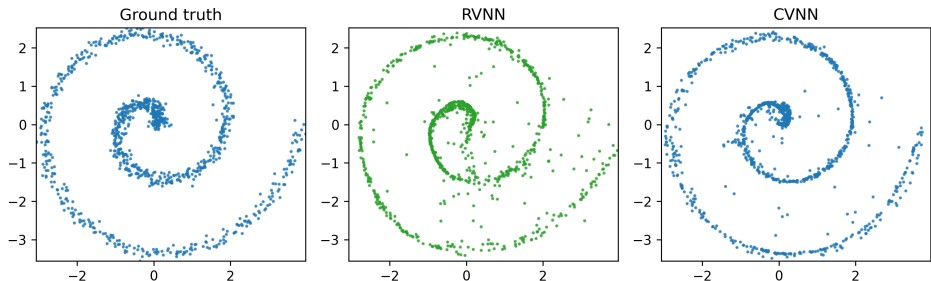

Figure 1: Ground-truth distribution compared with samples generated by RVNN and CVNN.

any explicit upsampling or autoregressive generation, thereby reducing latency. Early methods, such as the Griffin-Lim algorithm (Griffin & Lim, 1984), used iterative phase reconstruction but often yielded suboptimal coherence between magnitude and phase. GLA-Grad (Liu et al., 2024) later combined Griffin-Lim with neural diffusion models to improve phase accuracy.

More recent neural iSTFT-based vocoders, such as iSTFTNet (Kaneko et al., 2022), iSTFTNet2 (Kaneko et al., 2023), APNet (Ai & Ling, 2023), APNet2 (Du et al., 2024), FreeV (Lv et al., 2024), Vocos (Siuzdak, 2024), and RFWave (Liu et al., 2025), employ diverse architectural designs for iSTFT-based waveform generation.

In these systems, the STFT-domain coefficients are generated directly for frame-level synthesis, enabling efficient inference without waveform upsampling. Our work retains this benefit but additionally focuses on how this representation is modeled within the network. For this reason, we use complex-valued layers that operate directly in the complex domain rather than separating each coefficient into real and imaginary channels.

## 3 PRELIMINARY ANALYSIS OF REAL- AND COMPLEX-VALUED NETWORKS

Recent work on complex-valued neural networks suggests that operating directly in the complex field can better capture interactions between a variable's magnitude and phase than relying on real-valued parameterizations that treat the two components independently (Barrachina et al., 2021; Dou et al., 2025). Motivated by this perspective, we conduct a controlled generative experiment designed to isolate the effect of complex-domain modeling from architectural factors specific to waveform generation.

We train a lightweight MLP-based GAN on a synthetic complex distribution and compare two models: RVNN, which represents complex numbers as two real channels, and CVNN, which processes each coefficient as a single complex entity. Because the CVNN stores real and imaginary parameters separately, it requires roughly twice the memory for a given layer width; to match memory usage fairly, the RVNN is assigned twice the hidden dimension.

Table 1: JSD between the generated and ground-truth magnitude (mag.) and phase distributions for RVNN and CVNN.

| Model | JSD (mag.) | JSD (phase) |
|---|---|---|
| RVNN | $0.018350 \pm 0.014$ | $0.021110 \pm 0.036$ |
| CVNN | $0.006548 \pm 0.003$ | $0.003911 \pm 0.002$ |

Figure 1 presents sample visualizations across multiple training seeds, and Table 1 reports the Jensen–Shannon divergence (JSD) between the generated and target magnitude and phase distributions, computed using a kernel density–based estimator. Both models recover the broad structure of the target distribution, but the CVNN yields samples that adhere more closely to the underlying trajectory and exhibit lower JSD in both magnitude and phase.

These observations provide a simple, controlled example in which modeling directly in the complex domain offers representational advantages when the data possess inherent real–imaginary dependencies. This motivates our use of CVNNs in the proposed method that follows. Extended analysis and additional visualizations are included in the Appendix B.

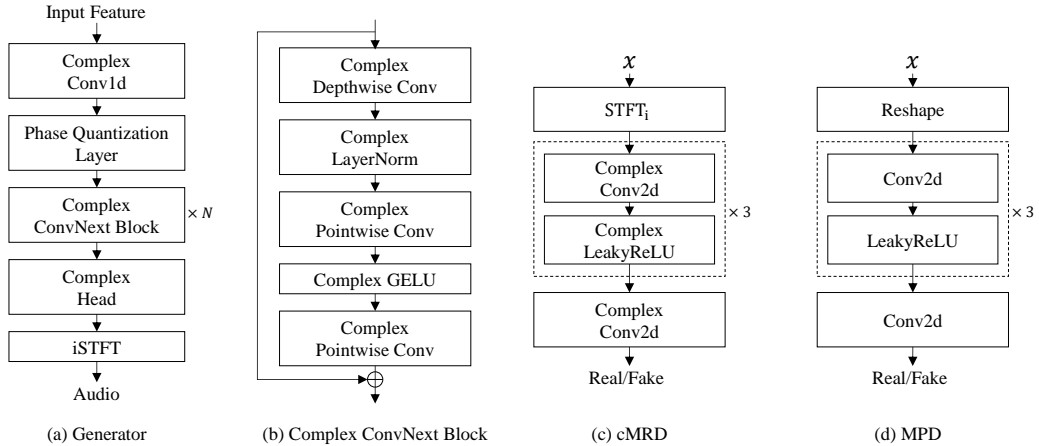

Figure 2: Overview of the ComVo architecture.

## 4 METHOD

We present ComVo, an iSTFT-based GAN vocoder whose generator and discriminator operate entirely in the complex domain, preserving real-imaginary interactions end to end. The model uses an iSTFT synthesis pipeline with adversarial training objectives. We also include a phase quantization layer as an inductive bias and adopt a block-matrix formulation for efficient complex-valued computation. Figure 2 provides an overview of the architecture.

### 4.1 GENERATOR

Figure 2(a) depicts our generator, which is adapted from the Vocos architecture (Siuzdak, 2024). We chose Vocos as our starting point because it synthesizes via frame-level iSTFT without requiring learned upsampling, features a compact feed-forward structure, and serves as a widely used baseline for comparison. All convolutions and normalizations in our generator are implemented in the complex domain. We use a split GELU activation (Hendrycks & Gimpel, 2016) to maintain the ConvNeXt-style block layout in the complex setting. After the initial complex convolution, a phase quantization layer discretizes phase values to stabilize training. Figure 2(b) details the complex ConvNeXt block used at each generator stage.

### 4.2 DISCRIMINATOR

We propose a complex multi-resolution discriminator (cMRD), as shown in Figure 2(c). Prior work on spectrogram-based discriminators typically used either only magnitude spectra or concatenated the real and imaginary spectrogram channels as independent inputs to a real-valued network (Jang et al., 2021; Siuzdak, 2024). In contrast, cMRD uses complex-valued layers and operates directly on complex spectrogram inputs. It comprises multiple sub-discriminators, each operating at a different STFT resolution. During training, we apply the adversarial loss separately to the real and imaginary parts. We also include a multi-period discriminator (MPD), shown in Figure 2(d), which consists of multiple sub-discriminators operating over different periods and processing reshaped waveform segments (Kong et al., 2020). Because the MPD operates at the waveform level, it remains a real-valued network. The overall training objective combines the adversarial losses from cMRD and MPD, along with feature matching and reconstruction losses. Full loss definitions and weights are provided in Appendix C.

### 4.3 PHASE QUANTIZATION LAYER

Complex-valued networks remain largely unexplored in terms of nonlinear transformations since any nonlinearity must jointly handle the real and imaginary components. We represent each Mel-spectrogram as a complex value by initializing the imaginary part to zero. We then introduce a phase quantization layer that discretizes phase angles into a fixed set of levels. This provides a structured

nonlinearity that preserves relative phase relationships and mitigates phase drift during training. For a complex feature $z = re^{i\theta}$, where $r \geq 0$ denotes the magnitude and $\theta \in (-\pi, \pi]$ denotes the principal phase, the quantized phase is defined as:

$$\theta_q = \frac{2\pi}{N_q} \cdot \text{round}\left(\frac{N_q}{2\pi}\theta\right),$$ (1)

where $N_q$ is the number of quantization levels. The quantized complex value is reconstructed as

$$z_q = re^{i\theta_q}.$$ (2)

Quantizing the phase by mapping continuous angles to a fixed set of levels introduces inherent discontinuities that would normally block gradient propagation. To preserve end-to-end differentiability, we adopt the straight-through estimator (STE) (Bengio, 2013), in which the quantization operation is applied in the forward pass, while its gradient is approximated by an identity function during backpropagation. This preserves gradient propagation through the phase quantization layer and improves optimization stability in practice. Furthermore, by restricting phase values to a discrete set, phase quantization acts as a form of regularization: it limits unwarranted phase variability in intermediate representations and guides the network toward learning more coherent and structured phase patterns.

## 4.4 OPTIMIZING COMPLEX COMPUTATION WITH BLOCK MATRICES

To improve efficiency in both the forward and backward passes, we reformulate CVNN operations as real-valued block-matrix multiplications. In many autodifferentiation systems, complex-valued layers are implemented by explicitly tracking real and imaginary components as separate real-valued tensors. This leads to redundant operations and inefficient memory access during both the forward and backward passes. We address this by adopting a block-wise formulation that represents complex values as structured pairs of real values and processes them jointly through unified matrix operations. This approach reduces component-wise operations and enhances parallelism on modern GPU architectures by enabling matrix-based execution throughout the computational graph. The forward complex operation can be expressed as:

$$\begin{bmatrix} \text{Re}(z') \\ \text{Im}(z') \end{bmatrix} = \begin{bmatrix} W_r & -W_i \\ W_i & W_r \end{bmatrix} \begin{bmatrix} x \\ y \end{bmatrix},$$ (3)

where $z = x + i\,y$ (with $x$ and $y$ denoting the real and imaginary input vectors), $W = W_r + i\,W_i$ is the complex weight matrix (with $W_r$, $W_i$ its real and imaginary parts), and $z'$ is the resulting complex output. The backward gradient computation uses the same block matrix structure:

$$\begin{bmatrix} \frac{\partial L}{\partial x} \\ \frac{\partial L}{\partial y} \end{bmatrix} = \begin{bmatrix} W_r & -W_i \\ W_i & W_r \end{bmatrix}^\top \begin{bmatrix} g_r \\ g_i \end{bmatrix},$$ (4)

where $g_r$ and $g_i$ are the real and imaginary components of the gradient from the next layer. This unified formulation is implemented for all parameterized CVNN layers via custom autograd functions. It reduces the number of separate operations and improves parallelism on GPUs by replacing four independent real-valued multiplies with a single block-matrix multiply, thereby eliminating redundant computation and allowing more efficient gradient evaluation.

## 5 RESULTS

### 5.1 EXPERIMENTAL SETUP

We train our model on the LibriTTS corpus (Zen et al., 2019), using the `train-clean-100`, `train-clean-360`, and `train-other-500` subsets for training, and evaluating on `test-clean` and `test-other` sets. All audio is sampled at 24 kHz. The STFT uses an FFT size of 1024, hop size of 256, and Hann window of length 1024. Mel-spectrograms are computed with 100 Mel-bins and a maximum frequency of 12 kHz. We compare ComVo against several representative vocoders: HiFi-GAN (v1) (Kong et al., 2020), iSTFTNet (Kaneko et al., 2022), BigVGAN

Table 2: Objective and subjective evaluation on the LibriTTS dataset.

| Model | UTMOS ↑ | MR-STFT ↓ | PESQ ↑ | Periodicity ↓ | V/UV F1 ↑ | MOS ↑ | CMOS ↑ |
|---|---|---|---|---|---|---|---|
| GT | 3.8712 | - | - | - | - | 4.08 ± 0.04 | 0.14 |
| HiFi-GAN | 3.3453 | 1.0455 | 2.9360 | 0.1554 | 0.9174 | 4.00 ± 0.05 | −0.09 |
| iSTFTNet | 3.3591 | 1.1046 | 2.8136 | 0.1476 | 0.9243 | 3.98 ± 0.05 | −0.04 |
| BigVGAN | 3.5197 | 0.8994 | 3.6122 | 0.1181 | 0.9418 | 4.05 ± 0.05 | −0.05 |
| Vocos | 3.6025 | 0.8856 | 3.6266 | 0.1061 | 0.9522 | 4.05 ± 0.05 | −0.02 |
| ComVo | **3.6901** | **0.8439** | **3.8239** | **0.0903** | **0.9609** | **4.07 ± 0.05** | 0 |

Table 3: Objective evaluation on the MUSDB18-HQ.

| Model | MR-STFT ↓ | PESQ ↑ | Periodicity ↓ | V/UV F1 ↑ |
|---|---|---|---|---|
| HiFi-GAN | 1.1909 | 2.3592 | 0.1804 | 0.9004 |
| iSTFTNet | 1.2388 | 2.2357 | 0.1815 | 0.9102 |
| BigVGAN | 0.9658 | 3.2391 | 0.1388 | 0.9340 |
| Vocos | 0.9307 | 3.2785 | 0.1369 | 0.9361 |
| ComVo | **0.8776** | **3.5220** | **0.1304** | **0.9384** |

Table 4: Subjective evaluation on the MUSDB18-HQ.

| Model | Vocals | Drums | Bass | Others | Mixture | Average |
|---|---|---|---|---|---|---|
| GT | 4.31 ± 0.11 | 4.25 ± 0.12 | 4.26 ± 0.12 | 4.29 ± 0.11 | 4.37 ± 0.11 | 4.29 ± 0.11 |
| HiFi-GAN | 3.83 ± 0.14 | 3.93 ± 0.13 | 3.43 ± 0.19 | 3.21 ± 0.19 | 3.60 ± 0.16 | 3.61 ± 0.16 |
| iSTFTNet | 3.82 ± 0.14 | 4.03 ± 0.13 | 3.37 ± 0.18 | 3.17 ± 0.19 | 3.52 ± 0.17 | 3.59 ± 0.17 |
| BigVGAN | **4.07 ± 0.12** | **4.19 ± 0.12** | 3.59 ± 0.17 | 3.57 ± 0.15 | 3.96 ± 0.12 | 3.88 ± 0.14 |
| Vocos | 4.04 ± 0.12 | 4.10 ± 0.13 | 3.58 ± 0.16 | 3.52 ± 0.17 | 3.87 ± 0.13 | 3.82 ± 0.14 |
| ComVo | 4.05 ± 0.12 | 4.14 ± 0.12 | **3.60 ± 0.17** | **3.68 ± 0.16** | **3.98 ± 0.13** | **3.89 ± 0.14** |

(base) (Lee et al., 2023), and Vocos (Siuzdak, 2024). For iSTFTNet, we use an open-source reimplementation, while the other models are trained using official code with recommended settings. We evaluate using both subjective and objective metrics. Subjective quality is assessed via mean opinion score (MOS), similarity MOS (SMOS), and comparison MOS (CMOS). Objective metrics include UTMOS (Saeki et al., 2022), PESQ (Rix et al., 2001), multi-resolution STFT (MR-STFT) error (Yamamoto et al., 2020), periodicity RMSE, and V/UV F1 score (Morrison et al., 2022). Detailed explanations are provided in Appendix K and Appendix L.

## 5.2 COMPARATIVE EVALUATION

Table 2 reports results on LibriTTS: ComVo achieves the highest objective scores among the baselines, and the corresponding MOS and CMOS are comparable to those of strong baseline systems. Tables 3 and 4 report results on MUSDB18-HQ (Rafii et al., 2019), an out-of-distribution audio dataset: ComVo achieves higher scores across all objective measures than the other models, and the corresponding subjective evaluations are comparable to strong baselines. The SMOS evaluation shows that ComVo delivers competitive perceptual quality across individual source stems and mixture tracks, with its average scores typically at or near the top. Taken together, these results indicate that an iSTFT-based model with complex-valued modeling consistently improves performance while maintaining the standard pipeline.

## 5.3 IMPACT OF COMPLEX-VALUED MODELING

We assess the contribution of each discriminator component individually. The MPD and MRD provide complementary forms of supervision: the MPD emphasizes periodic structure, while the MRD supplies multi-resolution spectral constraints. To understand how each behaves on its own, we evaluate MPD-only, MRD-only, and cMRD-only configurations. The MPD-only variant lacks spectral guidance and exhibits higher MR-STFT error. The MRD-only variant attains low STFT-based errors but produces a lower UTMOS score, indicating that spectral constraints alone do not fully capture

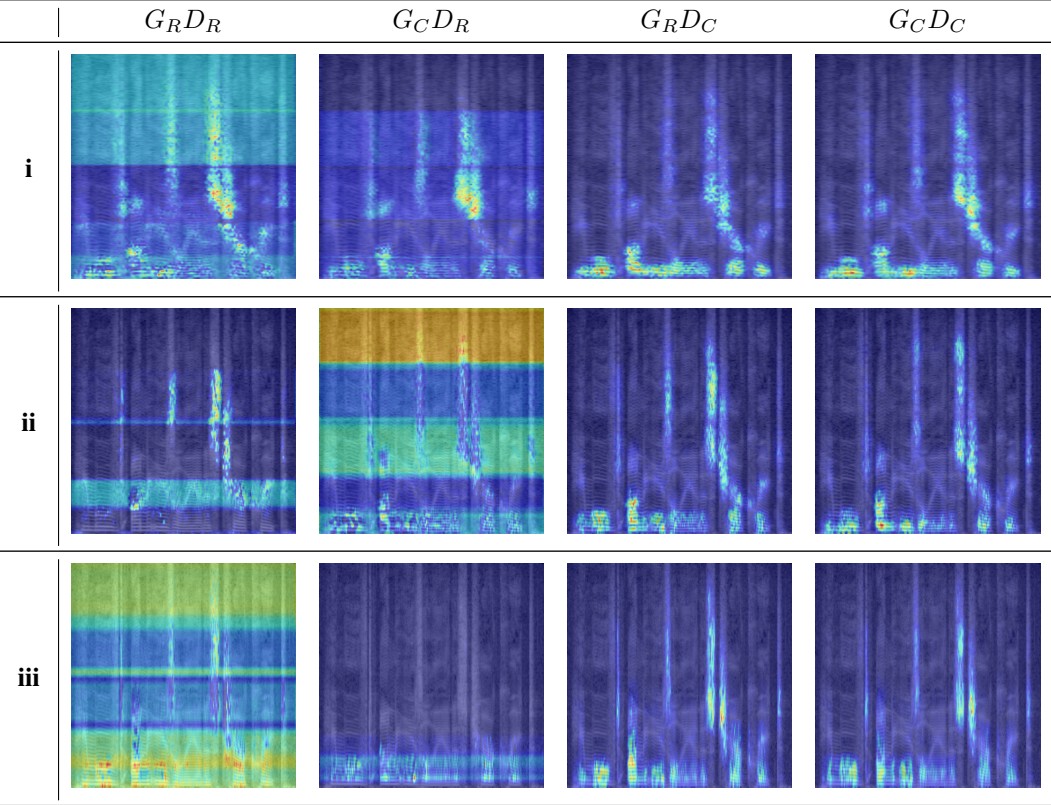

Figure 3: Grad-CAM comparison across generator-discriminator configurations. Each row corresponds to a cMRD sub-discriminator operating at a different STFT resolution (i, ii, iii).

perceptual quality. The cMRD-only model improves over the MRD-only baseline across all objective metrics, showing that the complex-valued discriminator provides a more effective constraint than its real-valued counterpart even when used alone.

We then extend the analysis to the full generator–discriminator combinations: $G_R D_R$, $G_C D_R$, $G_R D_C$, and $G_C D_C$, where $G_R$ and $G_C$ denote real-valued and complex-valued generators, and $D_R$ and $D_C$ denote real-valued and complex-valued discriminators. To isolate the effect of complex-valued modeling, the phase-quantization layer is disabled for all configurations, and the MPD branch is kept active without modification.

Replacing only the generator ($G_R D_R \rightarrow G_C D_R$) consistently improves all objective metrics. Replacing only the discriminator ($G_R D_R \rightarrow G_R D_C$) also yields measurable gains, particularly in MR-STFT error and PESQ. The best performance is achieved when both the generator and discriminator operate in the complex domain ($G_C D_C$), confirming the effectiveness of complex-domain modeling for iSTFT-based waveform generation.

For qualitative analysis, we visualize Grad-CAM (Selvaraju et al., 2017) activations of the discriminator in Figure 3. Each row in the figure corresponds to a sub-discriminator index (i, ii, iii), and each column corresponds to one of the generator-discriminator configurations. In the configurations with a real-valued MRD ($G_R D_R$ and $G_C D_R$), the attention maps are diffuse and poorly aligned with speech-relevant spectral structures. In contrast, in the configurations with a cMRD ($G_R D_C$ and $G_C D_C$), the highlighted regions consistently trace structured spectral patterns across all sub-discriminators. These results indicate that complex-valued discriminators provide more precise spectral feedback to the generator, helping it better match perceptually important features and ultimately improving synthesis quality, as also reflected in the ablation metrics.

Table 5: Ablation study comparing real-valued and complex-valued architectures.

| Model | UTMOS ↑ | MR-STFT ↓ | PESQ ↑ | Periodicity ↓ | V/UV F1 ↑ |
|---|---|---|---|---|---|
| MPD only | 3.6357 | 0.8522 | 3.7670 | 0.0942 | 0.9613 |
| MRD only | 2.8338 | 0.8442 | 3.9868 | 0.0870 | 0.9610 |
| cMRD only | 2.9285 | 0.8398 | 4.0149 | 0.0859 | 0.9635 |
| $G_R D_R$ | 3.6025 | 0.8856 | 3.6266 | 0.1061 | 0.9522 |
| $G_R D_C$ | 3.5930 | 0.8679 | 3.6399 | 0.1060 | 0.9497 |
| $G_C D_R$ | 3.6452 | 0.8597 | 3.7375 | 0.0978 | 0.9567 |
| $G_C D_C$ | 3.6646 | 0.8435 | 3.7756 | 0.0915 | 0.9625 |

Table 6: Ablation on phase quantization levels. $N_q$ denotes the number of quantization levels.

| $N_q$ Quantization | UTMOS ↑ | MR-STFT ↓ | PESQ ↑ | Periodicity ↓ | V/UV F1 ↑ |
|---|---|---|---|---|---|
| 0 | 3.6646 | **0.8435** | 3.7756 | 0.0915 | **0.9625** |
| 128 | **3.6901** | 0.8439 | 3.8239 | 0.0903 | 0.9609 |
| 256 | 3.6423 | 0.8466 | 3.8127 | 0.0926 | 0.9597 |
| 512 | 3.6412 | 0.8489 | **3.8248** | **0.0896** | 0.9613 |

Table 7: Comparison of standard PyTorch and refined implementations.

| Implementation | MR-STFT ↓ | GPU xRT ↑ | Training Time | Nodes (Gen / cMRD) |
|---|---|---|---|---|
| Native PyTorch | 0.8465 | **702.26** | 183 hrs | 5686 / 4248 |
| Block-matrix | **0.8435** | 696.91 | **138 hrs** | **2547 / 1404** |

## 5.4 EFFECT OF PHASE QUANTIZATION

Table 6 shows that adding a phase quantization layer yields clear benefits in perceptual quality, despite only a minor trade-off in reconstruction fidelity. The model without phase quantization ($N_q = 0$) achieves the lowest MR-STFT error, but a moderate quantization level ($N_q = 128$) smooths out phase fluctuations, resulting in higher UTMOS and PESQ scores and fewer periodicity artifacts, with only a small increase in MR-STFT error. Using finer quantization (e.g., $N_q = 256$, $N_q = 512$) can further boost perceptual metrics, but with diminishing returns and a slight degradation in reconstruction accuracy. Overall, phase quantization acts as an effective regularizer: it enhances listening quality while only modestly affecting spectral fidelity, with $N_q = 128$ providing the best trade-off in our setup.

## 5.5 BLOCK-MATRIX COMPUTATION SCHEME

In this section, we evaluate the efficiency and graph-complexity benefits of our block-matrix computation scheme. Table 7 reports the comparative results. It shows that our block-matrix implementation achieves performance comparable to PyTorch's native complex operations in terms of MR-STFT reconstruction error. While PyTorch's optimized complex kernels yield slightly faster forward-pass throughput, our overall training time is substantially shorter. Specifically, we reduce the number of backward graph nodes in the generator by over 55% and in the discriminator's cMRD by nearly 67%, resulting in a 25% reduction in training time. This improvement arises primarily from the backward pass: examining the gradient computation graphs reveals that our method dramatically lowers the node count compared to PyTorch's default approach of separately tracking real and imaginary components. By replacing four independent real-valued multiplications with a simple channel concatenation and a single matrix multiplication, we eliminate redundant operations and significantly accelerate gradient computation, all without sacrificing model fidelity.

## 5.6 EVALUATION IN TEXT-TO-SPEECH PIPELINE

We further evaluate each model in a text-to-speech (TTS) pipeline by pairing it with an acoustic model. In particular, we use Matcha-TTS (Mehta et al., 2024) as the acoustic model to generate Mel-spectrograms from text, then pass those spectrograms to each model. Matcha-TTS is trained on LibriTTS, and each model is trained independently on LibriTTS and connected to the Matcha-

Table 8: UTMOS, MOS, and CMOS comparison in the TTS pipeline.

| Model | UTMOS ↑ | MOS ↑ | CMOS ↑ |
|---|---|---|---|
| HiFi-GAN | 3.2233 | $3.85 \pm 0.05$ | $-0.22$ |
| iSTFTNet | 3.2951 | $3.89 \pm 0.05$ | $-0.15$ |
| BigVGAN | 3.3022 | $3.92 \pm 0.05$ | $-0.06$ |
| Vocos | 3.4357 | $3.91 \pm 0.05$ | $-0.06$ |
| ComVo | **3.4403** | $\textbf{3.92} \pm \textbf{0.05}$ | 0 |

Table 9: Comparison of computational cost and inference latency.

| Model | Param (M) | Memory (MB) | GPU xRT ↑ |
|---|---|---|---|
| HiFi-GAN | 14.00 | 53.40 | 259.08 |
| iSTFTNet | 13.33 | 50.83 | 402.21 |
| BigVGAN | 14.02 | 53.46 | 158.07 |
| Vocos | 13.54 | 51.62 | 4657.65 |
| ComVo | 13.28 | 101.24 | 819.02 |

Table 10: Objective evaluation and cost comparison: complex modeling vs. parameter scaling.

| Model | Params. (M) | Memory (MB) | UTMOS ↑ | MR-STFT ↓ | PESQ ↑ | Periodicity ↓ | V/UV F1 ↑ |
|---|---|---|---|---|---|---|---|
| $G_R D_R$ | 13.54 | 51.62 | 3.6025 | 0.8856 | 3.6266 | 0.1061 | 0.9522 |
| $G_R D_R$ 2× | 27.05 | 103.19 | 3.6164 | 0.8622 | 3.6336 | 0.1055 | 0.9524 |
| $G_C D_R$ | 13.28 | 101.24 | **3.6452** | **0.8597** | **3.7375** | **0.0978** | **0.9567** |

TTS outputs without additional fine-tuning. Table 8 reports the MOS, UTMOS, and CMOS for the TTS pipeline evaluation. ComVo achieves a MOS that matches the top score among the compared models, and it attains the highest UTMOS. This indicates that ComVo reliably converts the predicted spectrograms into high-quality waveforms within the TTS setting.

## 5.7 COMPUTATIONAL ANALYSIS

Table 9 compares the inference throughput and memory usage of each model under a common setup (batch size 1, no hardware-specific optimizations). HiFi-GAN and BigVGAN are upsampling-based models, whereas iSTFTNet, Vocos, and ComVo synthesize via frame-level iSTFT. The upsampling-based models exhibit the lowest throughput (lower xRT, indicating slower generation), while the iSTFT-based models run significantly faster. Among them, Vocos achieves the highest throughput. ComVo's throughput (xRT) lies within the range of the other iSTFT-based models. However, its memory footprint is higher than the real-valued iSTFT baselines: with a complex type, each weight is stored as a real–imaginary pair, so at the same precision the per-parameter memory is roughly doubled for a fixed parameter count.

To test whether the improvements stem merely from the larger memory footprint of complex types, we trained a real-valued model with twice the parameter count to match the complex model's memory and compared cost–quality trade-offs. The results are reported in Table 10. We compare three settings: the baseline real-valued model ($G_R D_R$), a widened real-valued model with roughly 2× parameters (denoted $G_R D_R$ 2×), and a complex-valued model ($G_C D_R$). The discriminator is identical across all settings. $G_C D_R$ and $G_R D_R$ 2× have comparable memory footprints. As expected, $G_R D_R$ 2× improves objective metrics relative to $G_R D_R$. In fact, $G_C D_R$ exceeds the widened model across all metrics despite a similar memory cost. Taken together, Tables 9 and 10 indicate that modeling real–imaginary correlations with CVNNs provides larger quality gains than simply scaling real-valued models.

## 6 LIMITATIONS

ComVo integrates complex-valued networks into an iSTFT-based vocoder. To keep the implementation straightforward, we adopt split-style designs. Concretely, we apply component-wise hinge losses to the real and imaginary outputs of cMRD, and we use split GELU within the ConvNeXt backbone. We will explore more advanced designs for these components in future work. The block-matrix formulation accelerates training, but computational overhead remains high because complex layers store and process paired real and imaginary values. Empirically, multi-GPU Distributed Data Parallel experiments showed under-optimized performance for complex parameters in our current training setup and occasional numerical issues; accordingly, we report single-GPU results. With better multi-GPU optimization and broader design exploration, larger-scale studies should be feasible and can further catalyze research on CVNNs for speech generation.

## 7 CONCLUSION

We presented ComVo, a vocoder that integrates CVNNs into both the generator and the discriminator, establishing a complex-domain adversarial framework for iSTFT-based waveform generation. By modeling the real and imaginary components jointly, our method addresses the structural mismatches in conventional real-valued processing of complex spectrograms. We also introduced a phase quantization layer as an inductive bias and a block-matrix formulation that simplifies computation graphs and accelerates training. ComVo delivered higher synthesis quality than comparable real-valued baselines. In addition, the block-matrix formulation reduced training time by approximately 25%. Future work will extend this framework beyond adversarial training to other generative paradigms (e.g., diffusion or flow-matching) and explore richer complex-domain activations and losses.

## ACKNOWLEDGMENTS

This work was partly supported by Institute of Information & communications Technology Planning & Evaluation(IITP) grant funded by the Korea government(MSIT) (No. RS-2019-II190079, Artificial Intelligence Graduate School Program (Korea University), IITP-2026-RS-2025-02304828, Artificial Intelligence Star Fellowship Support Program to nurture the best talents and No. RS-2024-00457882, AI Research Hub Project).

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

# A OVERVIEW OF COMPLEX-VALUED NEURAL NETWORKS

This section reviews the core building blocks of CVNNs—complex convolutions, activation functions, normalization, and optimization via Wirtinger calculus (Wirtinger, 1927). CVNNs extend real-valued networks by jointly modeling the real and imaginary components (Trabelsi et al., 2018). By preserving cross-component structure in the complex domain, they yield more coherent representations than split-channel parameterizations.

**Complex Convolutions**: A CVNN performs convolutions directly in the complex domain, jointly processing the real and imaginary parts. For an input complex feature $z = x + iy$ and a complex filter $h = a + ib$, the output $z'$ of a complex convolution is:

$$z' = (x * a - y * b) + i (x * b + y * a), \tag{5}$$

where $x, y$ are the real and imaginary components of $z$, and $a, b$ are the corresponding components of $h$. Here, $*$ denotes the convolution operation applied to each channel pair before recombining.

**Activation Functions**: Complex-valued networks require activation functions that handle both magnitude and phase in a coherent way. Let $f_{\mathrm{Re}}, f_{\mathrm{Im}}, f_{\mathrm{Mag}} : \mathbb{R} \to \mathbb{R}$ be real-valued nonlinearities. A simple split activation applies $f_{\mathrm{Re}}$ and $f_{\mathrm{Im}}$ separately to the real and imaginary components:

$$f(z) = f_{\mathrm{Re}}(x) + i \, f_{\mathrm{Im}}(y), \tag{6}$$

but this approach ignores the natural coupling between magnitude and phase. A more phase-aware alternative applies $f_{\mathrm{Mag}}$ to the magnitude and then reattaches the original phase:

$$f(z) = f_{\mathrm{Mag}}(|z|) \, e^{i\theta}, \tag{7}$$

thereby preserving all phase information while still introducing the desired nonlinearity. (Here $|z|$ is the magnitude and $\theta$ is the phase of $z = re^{i\theta}$.)

**Normalization**: Normalization in CVNNs accounts for the joint distribution of real and imaginary components. A general form of complex normalization is:

$$z_{\mathrm{norm}} = \frac{z - \mu}{\sigma}, \tag{8}$$

where $\mu$ and $\sigma$ are the mean and standard deviation of the complex input. To capture correlations between the real and imaginary parts, this basic normalization is extended using the covariance matrix:

$$\Sigma = \begin{bmatrix} \sigma_{xx} & \sigma_{xy} \\ \sigma_{yx} & \sigma_{yy} \end{bmatrix}, \tag{9}$$

where $\sigma_{xx}$ and $\sigma_{yy}$ denote the variances of the real and imaginary components, respectively, and $\sigma_{xy} = \sigma_{yx}$ represents their cross-covariance. Using the estimated covariance, the input is normalized by centering and decorrelating:

$$z_{\mathrm{norm}} = \Sigma^{-1/2}(z - \mu), \tag{10}$$

and an affine transformation is then applied to restore the network's ability to shift and scale the normalized features:

$$z' = \gamma z_{\mathrm{norm}} + \beta, \tag{11}$$

where $\gamma$ and $\beta$ are learnable complex-valued parameters. This formulation can be applied to various normalizations (e.g., layer or instance normalization) while preserving the complex structure.

**Gradient Optimization**: Gradient computation in CVNNs requires special care due to the non-holomorphic nature of most complex-valued functions. To handle this, CVNNs employ Wirtinger calculus (Wirtinger, 1927), which defines the gradient of a real-valued loss $L(z)$ with respect to a complex variable $z = x + iy$ as:

$$\frac{\partial L}{\partial z} = \frac{1}{2} \left( \frac{\partial L}{\partial x} - i \frac{\partial L}{\partial y} \right), \quad \frac{\partial L}{\partial \bar{z}} = \frac{1}{2} \left( \frac{\partial L}{\partial x} + i \frac{\partial L}{\partial y} \right). \tag{12}$$

For real-valued objectives, only the conjugate gradient $\frac{\partial L}{\partial \bar{z}}$ is used for parameter updates, which ensures descent in the loss landscape:

$$z^{(t+1)} = z^{(t)} - \eta \frac{\partial L}{\partial \bar{z}}, \tag{13}$$

where $\eta$ is the learning rate.

Table 11: Architecture used for both the CVNN and RVNN generators and discriminators. The two networks share the same layer structure and differ only in how complex variables are represented.

| Component | CVNN | RVNN |
|---|---|---|
| Input | $\mathbb{C}^1$ | $\mathbb{R}^2$ |
| Hidden dimension | 128 | 256 |
| Depth | 4 | 4 |
| Layer type | Complex Linear | Linear |
| Activation | Complex LeakyReLU | LeakyReLU |
| Output | $\mathbb{C}^1$ | $\mathbb{R}^2$ |

## B    INVESTIGATING REAL AND COMPLEX MODELS FOR COMPLEX-DOMAIN GENERATION

To investigate how real-valued and complex-valued neural networks differ in learning distributions with coupled real–imaginary structure, we conduct a minimal generative modeling experiment based on a two-dimensional target density defined in the complex plane. This setting removes the influence of architectural factors specific to waveform generation and isolates the effect of the underlying parameterization. The target distribution contains a nontrivial correlation between its real and imaginary components, providing a simple but informative test case for comparing representational behavior.

The complex-valued models operate directly in $\mathbb{C}$, receiving a one-dimensional complex latent variable and propagating it through a stack of complex linear layers with complex activations. The real-valued models use an equivalent architecture in depth but operate entirely in $\mathbb{R}$, starting from a two-dimensional latent input and producing two real outputs that are interpreted as the real and imaginary components of a sample. To match representational width between the two model families, the hidden dimension of the RVNN layers is doubled relative to the CVNN. A concise summary of these architectural differences is provided in Table 11. All networks are trained using the standard GAN objective with binary cross-entropy loss and identical optimization hyperparameters.

For each random seed, we examine three aspects of the learned distribution: (i) the scatter plot of generated samples, (ii) the magnitude histogram, and (iii) the phase histogram. These visualizations allow us to assess how consistently each model reproduces the target structure across independent runs. Examples are shown in Figure 4. While both models are capable of approximating the global geometry of the target, the complex-valued generator often produces more stable spirals and magnitude–phase statistics with reduced run-to-run variability. This experimental design does not aim to assert broad conclusions beyond this setting, but it provides a controlled example in which complex-valued parameterization can yield advantages when the modeled data are inherently expressed in the complex domain.

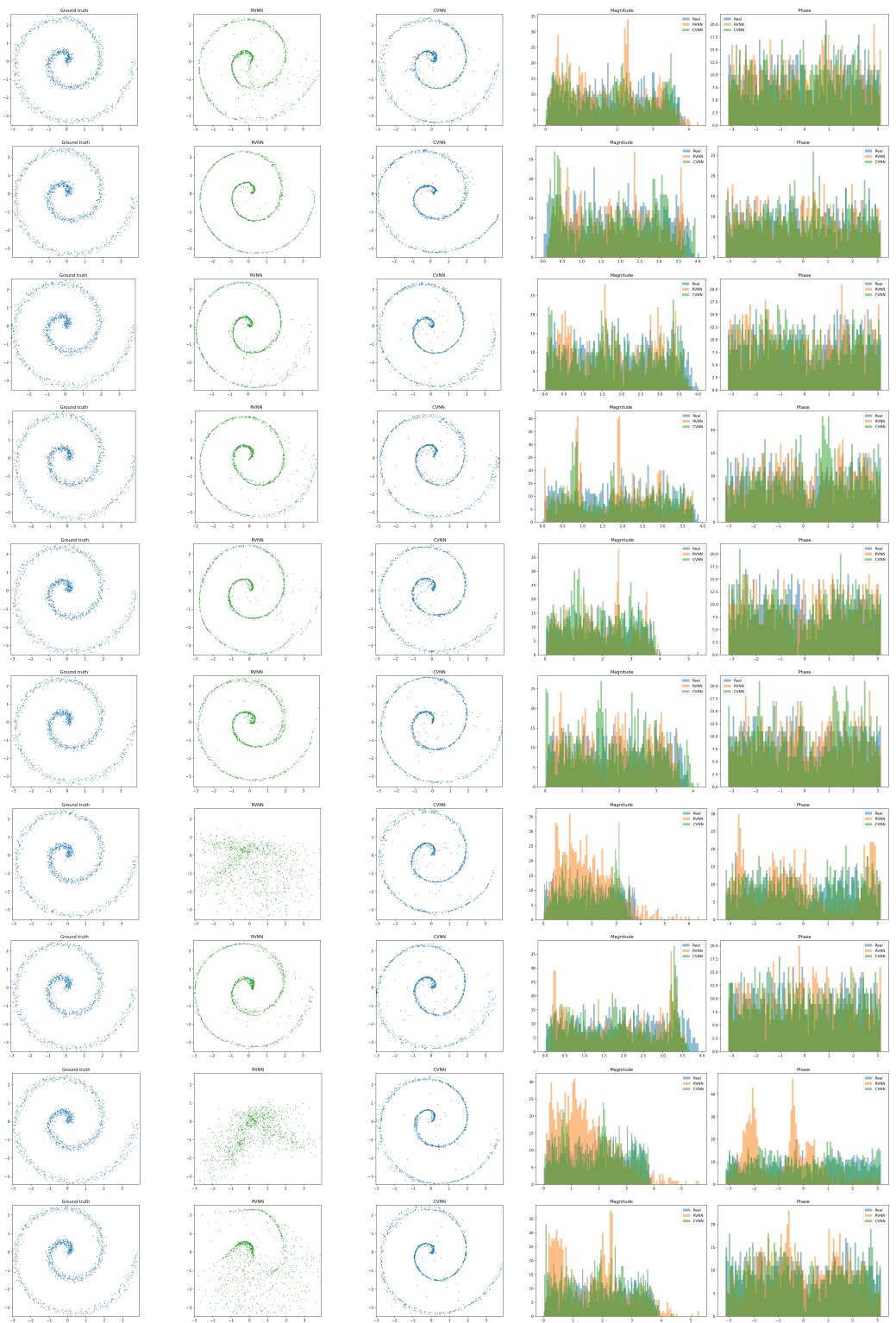

Figure 4: Visualizations over multiple training seeds. Each row corresponds to one run and contains five subplots: ground-truth samples, RVNN outputs, CVNN outputs, and the corresponding magnitude and phase distributions. This layout enables a run-to-run comparison of distributional behavior across the two models.

## C  DETAILS OF TRAINING OBJECTIVE

The ComVo training objective integrates adversarial, reconstruction, and feature-matching losses from both the MPD and the cMRD.

### C.1  DISCRIMINATOR LOSS

We use adversarial losses to push real samples above and generated samples below the decision boundary.

**MPD Loss**: Let $D_k^{\text{MPD}}$ denote the $k$-th sub-discriminator operating on raw waveforms. For each period $P_k$, the input segment $y$ is reshaped to $(P_k, T/P_k)$ to expose the periodic structure. We use a hinge loss on the real-valued outputs:

$$\mathcal{L}_D^{\text{MPD}} = \sum_{k=1}^{K} \Big[ \mathbb{E}_y \big( \max(0,\, 1 - D_k^{\text{MPD}}(y)) \big)$$
$$+ \ \mathbb{E}_{\hat{y}} \big( \max(0,\, 1 + D_k^{\text{MPD}}(\hat{y})) \big) \Big], \tag{14}$$

where $y$ and $\hat{y}$ are ground-truth and generated waveform segments, respectively.

**cMRD Loss**: We apply hinge losses independently to the real and imaginary components to retain compatibility with standard real-valued GAN objectives, while allowing the discriminator to operate directly in the complex domain. For any complex quantity $u$, let $[u]_R$ and $[u]_I$ denote its real and imaginary parts, respectively (these are operators on a single complex output, not separate networks). With $D_k^{\text{cMRD}}$ the $k$-th sub-discriminator,

$$\mathcal{L}_D^{\text{cMRD}} = \sum_{k=1}^{K} \Big[ \tfrac{1}{2} \mathbb{E}_z \big( \max(0,\, 1 - [D_k^{\text{cMRD}}(z)]_R) + \max(0,\, 1 - [D_k^{\text{cMRD}}(z)]_I) \big)$$
$$+ \ \tfrac{1}{2} \mathbb{E}_{\hat{z}} \big( \max(0,\, 1 + [D_k^{\text{cMRD}}(\hat{z})]_R) + \max(0,\, 1 + [D_k^{\text{cMRD}}(\hat{z})]_I) \big) \Big]. \tag{15}$$

### C.2  GENERATOR LOSS

The generator objective includes reconstruction, adversarial, and feature-matching terms.

**Mel-spectrogram Loss**: We use an $L_1$ loss on log-scaled Mel-spectrograms:

$$\mathcal{L}_{\text{Mel}} = \mathbb{E} \big\| M(y) - M(\hat{y}) \big\|_1, \tag{16}$$

where $y$ and $\hat{y}$ denote ground-truth and generated waveforms, and $M(\cdot)$ is the log-Mel transform.

**Adversarial Generator Loss**: For the MPD operating on waveform segments $\hat{y}$:

$$\mathcal{L}_G^{\text{MPD}} = \sum_{k=1}^{K} \mathbb{E}_{\hat{y}} \big( \max(0,\, 1 - D_k^{\text{MPD}}(\hat{y})) \big). \tag{17}$$

For the cMRD operating on generated spectrograms $\hat{z}$, let $[\,\cdot\,]_R$ and $[\,\cdot\,]_I$ denote the real and imaginary parts of a complex output. We apply hinge losses to both components:

$$\mathcal{L}_G^{\text{cMRD}} = \sum_{k=1}^{K} \tfrac{1}{2} \mathbb{E}_{\hat{z}} \Big( \max(0,\, 1 - [D_k^{\text{cMRD}}(\hat{z})]_R) + \max(0,\, 1 - [D_k^{\text{cMRD}}(\hat{z})]_I) \Big). \tag{18}$$

**Feature Matching Loss**: We match intermediate representations in both discriminators.

For MPD (waveform segments $y$ and $\hat{y}$), we use an $\ell_1$ loss on feature maps:

$$\mathcal{L}_{\text{FM}}^{\text{MPD}} = \sum_{k=1}^{K} \sum_{l=1}^{L_k} \mathbb{E} \big\| D_{k,l}^{\text{MPD}}(y) - D_{k,l}^{\text{MPD}}(\hat{y}) \big\|_1, \tag{19}$$

where $D_{k,l}^{\mathrm{MPD}}$ is the $l$-th layer feature of the $k$-th MPD sub-discriminator.

For cMRD (complex spectrograms $z$ and $\hat{z}$), let $[\,\cdot\,]_R$ and $[\,\cdot\,]_I$ denote the real and imaginary parts of a complex feature, respectively. We match the components separately:

$$
\mathcal{L}_{\mathrm{FM}}^{\mathrm{cMRD}} = \sum_{k=1}^{K} \sum_{l=1}^{L_k} \tfrac{1}{2}\, \mathbb{E}\Big( \big\| [D_{k,l}^{\mathrm{cMRD}}(z)]_R - [D_{k,l}^{\mathrm{cMRD}}(\hat{z})]_R \big\|_1 \tag{20}
$$
$$
+ \big\| [D_{k,l}^{\mathrm{cMRD}}(z)]_I - [D_{k,l}^{\mathrm{cMRD}}(\hat{z})]_I \big\|_1 \Big).
$$

**Total Generator Loss**: The generator objective combines reconstruction, adversarial, and feature-matching terms:

$$
\mathcal{L}_{\mathrm{gen}} = \lambda_{\mathrm{Mel}}\, \mathcal{L}_{\mathrm{Mel}} \; + \; \lambda_{\mathrm{MPD}} \big( \mathcal{L}_G^{\mathrm{MPD}} + \mathcal{L}_{\mathrm{FM}}^{\mathrm{MPD}} \big) \tag{21}
$$
$$
+ \; \lambda_{\mathrm{cMRD}} \big( \mathcal{L}_G^{\mathrm{cMRD}} + \mathcal{L}_{\mathrm{FM}}^{\mathrm{cMRD}} \big).
$$

Here, $\lambda_{\mathrm{Mel}}$, $\lambda_{\mathrm{MPD}}$, and $\lambda_{\mathrm{cMRD}}$ weight the Mel, MPD, and cMRD terms, respectively. Detailed hyperparameters are provided in Table 20.

# D  PROOF OF EQUIVALENCE BETWEEN THE BLOCK-MATRIX COMPUTATION SCHEME AND STANDARD COMPLEX-VALUED OPERATIONS

We now verify in detail that applying the block-matrix operator

$$
A = \begin{bmatrix} W_r & -W_i \\ W_i & W_r \end{bmatrix}
$$

to the stacked real vector $\begin{bmatrix} x;\, y \end{bmatrix}$ reproduces exactly the real and imaginary components of the complex product $z' = W z$ with $W = W_r + i\, W_i$.

## D.1  FORWARD COMPUTATION

Let
$$
z = x + i\, y, \quad W = W_r + i\, W_i,
$$
where $x, y, W_r, W_i$ are real-valued. Then the complex linear transformation can be written as
$$
\begin{aligned}
W z &= (W_r + iW_i)(x + i\, y) \\
&= W_r x + i\, W_i x + i\, W_r y + i^2\, W_i y \\
&= (W_r x - W_i y) \; + \; i\, (W_i x + W_r y).
\end{aligned}
$$
Thus
$$
\mathrm{Re}(z') = W_r x - W_i y, \quad \mathrm{Im}(z') = W_i x + W_r y.
$$
On the other hand, the block-matrix product gives
$$
A \begin{bmatrix} x \\ y \end{bmatrix} = \begin{bmatrix} W_r x - W_i y \\ W_i x + W_r y \end{bmatrix} = \begin{bmatrix} \mathrm{Re}(z') \\ \mathrm{Im}(z') \end{bmatrix}.
$$

## D.2  BACKWARD COMPUTATION

Let the scalar loss be $L$, and denote
$$
g_r = \frac{\partial L}{\partial \mathrm{Re}(z')}, \quad g_i = \frac{\partial L}{\partial \mathrm{Im}(z')}.
$$
In the complex formulation, the gradient with respect to $z$ is
$$
\frac{\partial L}{\partial z} = W^H (g_r + i\, g_i) = \big( W_r^\top g_r + W_i^\top g_i \big) + i\, \big( -W_i^\top g_r + W_r^\top g_i \big).
$$

Define

$$g_x = W_r^\top g_r + W_i^\top g_i, \quad g_y = -W_i^\top g_r + W_r^\top g_i.$$

Stacking these gives

$$\begin{bmatrix} g_x \\ g_y \end{bmatrix} = A^\top \begin{bmatrix} g_r \\ g_i \end{bmatrix} = \begin{bmatrix} W_r & -W_i \\ W_i & W_r \end{bmatrix}^\top \begin{bmatrix} g_r \\ g_i \end{bmatrix}, \tag{22}$$

which is precisely the transpose of the forward block-matrix. For convolutional layers, each transpose block corresponds to the appropriate transposed-convolution operator.

Table 12: Average GPU execution times for generator (Gen) and discriminator (Disc) forward and backward passes.

| Implementation | Gen Forward (s) | Gen Backward (s) | Disc Forward (s) | Disc Backward (s) |
|---|---|---|---|---|
| Native PyTorch | 0.005288 | 0.234591 | 0.079073 | 0.190067 |
| Gaussian trick | 0.005160 | 0.231545 | 0.074103 | 0.184894 |
| Block-matrix | 0.005786 | 0.181283 | 0.050389 | 0.139159 |

## E    SPEED COMPARISON OF GENERATOR AND DISCRIMINATOR OPERATIONS

To isolate the effect of block-matrix fusion, we benchmark only the generator and the cMRD, excluding the MPD and reusing the same pretrained hyperparameters across all implementations.

In addition to the native PyTorch implementation and our block-matrix formulation, we also evaluated Gauss' multiplication trick, implemented using the `complextorch` library[1].

Gauss' multiplication trick rewrites a complex product using three real-valued convolutions instead of four, and is a common arithmetic reduction technique for complex operations.

Table 12 reports the average GPU execution times for the forward and backward passes of both the generator and the cMRD over 10 runs with a batch size of 16. For the generator, the forward time shows minimal variation across implementations, indicating that fusing real and imaginary components introduces little overhead in this part of the computation. In contrast, the block-matrix formulation substantially reduces the generator's backward time and provides clear improvements in both the forward and backward passes of the cMRD, leading to a noticeably faster end-to-end training step. Overall, these results indicate that the block-matrix formulation can provide practical efficiency gains in our training setup.

Table 13: Component-level differences in intermediate values and parameter gradients between native and refined implementations.

| Metric | Conv1d | Conv2d | Linear |
|---|---|---|---|
| Input gradient | 2e-09 | 6e-09 | 8e-10 |
| Forward output | 4e-09 | 1e-08 | 7e-09 |
| Forward output gradient | 9e-09 | 2e-08 | 1e-08 |
| Weight | 0e+00 | 0e+00 | 0e+00 |
| Weight gradient | 1e-07 | 4e-07 | 4e-08 |
| Bias | 0e+00 | 0e+00 | 0e+00 |
| Bias gradient | 6e-08 | 4e-07 | 3e-08 |

## F    NUMERICAL CONSISTENCY VERIFICATION

To confirm that our block-matrix computation scheme maintains numerical fidelity, we compare forward outputs and gradients for each module against the native PyTorch implementation. Table 13

---

[1]https://github.com/josiahwsmith10/complextorch

Table 14: Model-level differences in outputs, losses, and gradient magnitudes between native and refined implementations.

| Metric | Generator | Discriminator |
|---|---|---|
| Forward output | 7e-06 | 5e-06 |
| Loss | 5e-07 | 2e-07 |
| Gradient | 5e-06 | 1e-06 |

reports mean absolute differences at the layer level for convolutional and linear modules—all within typical floating-point tolerances ($\sim 10^{-7}$). Table 14 summarizes end-to-end deviations in generator and discriminator outputs, losses, and gradient norms, all below $10^{-5}$. These results verify that, despite the structural optimizations, our block-matrix approach preserves numerical consistency and does not affect training dynamics.

## G  BACKWARD GRAPH VISUALIZATION

Figures 9, 10, and 11 show the backward computation graphs of the generator using (i) the native PyTorch complex implementation, (ii) Gauss' multiplication trick, and (iii) the block-matrix formulation, respectively. Figures 12, 13, and 14 present the corresponding graphs for the cMRD. For clarity, both models are simplified by using a single Mel-spectrogram loss and reducing the number of layers and channels.

Across all configurations, the block-matrix formulation (Figures 11 and 14) yields the most compact backward graph. Compared to the native (Figures 9, 12) and Gauss-based implementations (Figures 10, 13), it avoids redundant branches and reduces the number of elementwise operations, resulting in a significantly simpler and more efficient gradient flow.

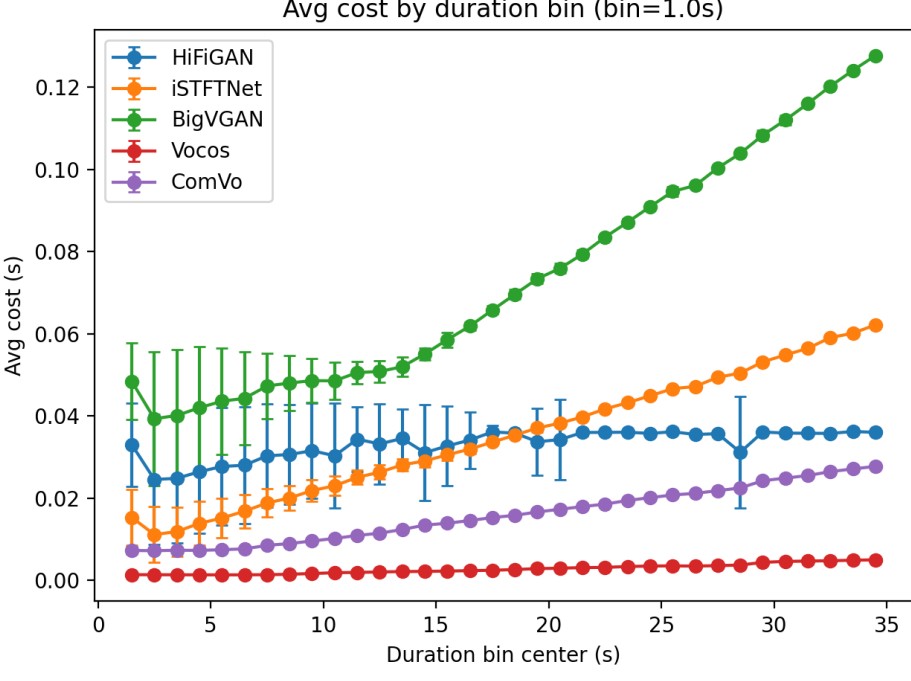

Figure 5: Average inference cost as a function of utterance duration.

## H  RUNTIME AS A FUNCTION OF UTTERANCE LENGTH

Figure 5 plots average inference cost versus utterance duration using 1-second bins and consistency under the same setup as Table 9; points indicate bin means and vertical bars show variability. Upsampling-based vocoders increase approximately in proportion to duration with a clear positive slope, whereas iSTFT-based vocoders exhibit a flatter, near-constant profile over the plotted range. The proposed method follows the iSTFT family: its curve lies above Vocos but remains below iSTFTNet and the upsampling-based systems across bins. Although CVNNs introduce computational overhead, ComVo maintains competitive runtime characteristics within the iSTFT class.

Table 15: Ablation comparing real Conv1D producing a two-channel (Re, Im) feature, Complex Conv w/o PQ, and Complex Conv w/ PQ

| Model | UTMOS ↑ | MR-STFT ↓ | PESQ ↑ | Periodicity ↓ | V/UV F1 ↑ |
|---|---|---|---|---|---|
| Real Conv | 3.6337 | 0.8610 | 3.7774 | 0.0980 | 0.9574 |
| Complex Conv w/o PQ | 3.6646 | 0.8435 | 3.7756 | 0.0915 | 0.9625 |
| Complex Conv w/ PQ | **3.6901** | **0.8439** | **3.8239** | **0.0903** | **0.9609** |

## I  ANALYSIS OF PHASE QUANTIZATION

The generator receives real-valued inputs, and the imaginary component of the initial complex representation must therefore be synthesized internally by the network. At this early stage, the phase can vary freely, as there is no signal-driven constraint guiding how the initial complex feature should be formed. Prior work in speech coding has also observed that unconstrained phase can introduce instability or unnecessary variability during optimization (Yu & Chan, 2002; Kim, 2003). Motivated by these considerations, we insert a phase quantization (PQ) step immediately after the first complex Conv1D layer to lightly regularize the formation of the initial complex features while allowing subsequent layers to operate without explicit phase constraints.

To examine whether the effect of PQ is tied to its interaction with the first complex layer, we trained a variant where the first complex Conv1D was replaced with a real Conv1D that outputs two channels, which are then interpreted as the real and imaginary components of a complex feature. Aside from this modification, the architecture remains unchanged. This variant trains properly and produces results similar to the version without PQ, whereas the original configuration with a complex Conv1D followed by PQ achieves higher scores across all metrics (Table 15). This comparison indicates that the benefit of PQ is associated with its placement at the point where complex features first emerge.

Table 16: Comparison with amplitude–phase prediction vocoders

| Model | UTMOS ↑ | MR-STFT ↓ | PESQ ↑ | Periodicity ↓ | V/UV F1 ↑ |
|---|---|---|---|---|---|
| APNet | 2.4015 | 1.3375 | 2.8457 | 0.1582 | 0.9185 |
| APNet2 | 2.7379 | 1.1582 | 2.7748 | 0.1448 | 0.9243 |
| FreeV | 2.6971 | 1.1782 | 2.7960 | 0.1581 | 0.9105 |
| ComVo | 3.6901 | 0.8439 | 3.8239 | 0.0903 | 0.9609 |

## J  COMPARISON WITH AMPLITUDE–PHASE PREDICTION VOCODERS

In addition to comparing against GAN-based vocoders, we also consider amplitude–phase prediction methods, in which magnitude and phase are modeled separately using real-valued networks. Representative examples include APNet (Ai & Ling, 2023), APNet2 (Du et al., 2024), and FreeV (Lv et al., 2024), all of which treat the two components as independent regression targets.

To position our complex-domain formulation relative to this family of methods, we trained APNet, APNet2, and FreeV with their official implementations under the same data and training settings used in our system. This provides a controlled comparison between explicit amplitude–phase estimation and directly modeling complex STFT coefficients.

Table 16 presents the results. Across all metrics, the proposed model achieves higher quality, suggesting that learning in the complex domain is an effective parameterization for iSTFT-based generation compared to treating magnitude and phase as separate prediction targets.

Table 17: Baseline model implementations and sources.

| Model | Implementation Source |
|---|---|
| iSTFTNet | https://github.com/rishikksh20/iSTFTNet-pytorch |
| HiFi-GAN | https://github.com/jik876/hifi-gan |
| BigVGAN | https://github.com/NVIDIA/BigVGAN |
| Vocos | https://github.com/gemelo-ai/vocos |
| APNet | https://github.com/YangAi520/APNet |
| APNet2 | https://github.com/redmist328/APNet2 |
| FreeV | https://github.com/BakerBunker/FreeV |

## K  BASELINE MODEL IMPLEMENTATIONS

We evaluate our proposed method against several representative neural vocoders, each with distinct architectural designs:

**HiFi-GAN (v1)** (Kong et al., 2020): A GAN-based vocoder that uses multiple discriminators (MPD and MRD) with a transposed convolutional generator. It emphasizes high-fidelity waveform generation with fast inference.

**iSTFTNet** (Kaneko et al., 2022): A lightweight vocoder that replaces upsampling layers with iSTFT to reduce redundant computations. It directly predicts complex-valued spectrograms, simplifying the overall architecture.

**BigVGAN (base)** (Lee et al., 2023): An improved HiFi-GAN variant that introduces the Snake function (Ziyin et al., 2020) for better modeling of periodicity and high-frequency details. It also adopts a scaled discriminator design, contributing to more stable GAN training and enhanced performance on challenging inputs.

**Vocos** (Siuzdak, 2024): An iSTFT-based vocoder built on a ConvNeXt (Liu et al., 2022) architecture that predicts Fourier spectral coefficients for waveform reconstruction. It achieves high-quality synthesis with low latency.

**APNet** (Ai & Ling, 2023): A vocoder that separately predicts amplitude and phase spectra using independent real-valued branches. Phase is modeled explicitly through a parallel estimation module with anti-wrapping losses, and the waveform is reconstructed via iSTFT.

**APNet2** (Du et al., 2024): An improved version of APNet that adopts a ConvNeXt v2 backbone and multi-resolution discriminators. It retains the separate amplitude–phase prediction design while offering higher fidelity and greater training stability.

**FreeV** (Lv et al., 2024): A lightweight amplitude–phase vocoder derived from APNet2 that incorporates signal-processing priors. It obtains an approximate amplitude spectrum via pseudo-inverse mel filtering, reducing ASP complexity while maintaining quality.

We use the official implementations provided by the authors whenever available, except for iSTFTNet, which lacks an official repository. For iSTFTNet, we adopt a publicly available open-source implementation instead. Implementation sources are summarized in Table 17.

Table 18: Implementation sources for objective evaluation metrics.

| Model | Implementation Source |
|---|---|
| UTMOS | https://github.com/sarulab-speech/UTMOS22 |
| MR-STFT | https://github.com/csteinmetz1/auraloss |
| PESQ | https://github.com/ludlows/PESQ |
| Periodicity RMSE & V/UV F1 score | https://github.com/descriptinc/cargan |

## L    EVALUATION METRICS

### L.1    SUBJECTIVE EVALUATION

We conducted mean opinion score (MOS) listening tests on Mechanical Turk with 20 U.S.-based native English speakers, each evaluating 50 samples. We also ran similarity mean opinion score (SMOS) tests under the same conditions. In MOS, listeners rated naturalness on a 1–5 scale; in SMOS, they rated similarity between synthesized and reference audio on a 1–5 scale. In addition, we conducted comparison MOS (CMOS) using a 7-point scale. For reporting, we use pairwise comparisons against our system as the reference; thus the reference row is centered at 0 and other systems' scores reflect average preference relative to it. To filter inattentive participants, we inserted fake samples and instructed listeners to mark them as "X"; any listener who missed these was excluded. Figure 6 shows the MOS interface, Figure 7 shows the SMOS interface and Figure 8 shows the CMOS interface.

### L.2    OBJECTIVE EVALUATION

We measure performance using five objective metrics: UTMOS (Saeki et al., 2022), multi-resolution short-time Fourier transform error (MR-STFT) (Yamamoto et al., 2020), perceptual evaluation of speech quality (PESQ) (Rix et al., 2001), periodicity RMSE, and voiced/unvoiced (V/UV) F1 score (Morrison et al., 2022). Implementation sources are listed in Table 18.

**UTMOS**: We use the open-source UTMOS model to predict MOS scores for evaluating speech naturalness.

**MR-STFT**: We use the multi-resolution STFT loss implementation from Auraloss (Steinmetz & Reiss, 2020) to measure spectral distortion between the generated and ground-truth audio.

**PESQ**: We use the wideband version of PESQ with audio resampled to 16 kHz to assess perceptual quality.

**Periodicity and V/UV F1**: Periodicity RMSE is used to quantify periodic artifacts, while the V/UV F1 score measures the accuracy of voiced/unvoiced classification.

Table 19: Comparison of large-scale models

| Model | Params. (M) | UTMOS ↑ | MR-STFT ↓ | PESQ ↑ | Periodicity ↓ | V/UV F1 ↑ |
|---|---|---|---|---|---|---|
| GT | - | 3.8712 | - | - | - | - |
| BigVGAN (large) | 112.41 | 3.5489 | 0.8644 | 3.8197 | 0.0888 | 0.9607 |
| Vocos (large) | 114.51 | 3.6923 | 0.8625 | 3.8362 | 0.0933 | 0.9596 |
| ComVo (large) | 114.56 | **3.7337** | **0.8443** | **3.8831** | **0.0871** | **0.9629** |

## M    EXTENDED EXPERIMENTS WITH LARGE-SCALE CONFIGURATIONS

To test whether the benefits of complex-valued modeling persist at higher capacity, we conducted a scaling study with large variants of the baselines and our model. All systems were trained on the same LibriTTS splits as in the base-scale experiments. For BigVGAN, we used the authors' official large configuration; for Vocos and ComVo, we set configurations to match the BigVGAN large model's parameter budget as closely as possible while keeping architectures comparable. All runs were trained for 1M optimization steps on a single GPU. Table 19 summarizes the large-scale results. In this setting, ComVo scaled effectively, showing clear quality gains across evaluation metrics. Overall, the complex-valued approach scales well, and increasing capacity yields consistent quality gains.

Table 20: Training hyperparameters.

| Mel-spectrogram | | |
|---|---|---|
| Sampling rate | | 24,000 |
| FFT size | | 1024 |
| Hop length | | 256 |
| Window size | | 1024 |
| Mel bins | | 100 |

| Generator | Base | Large |
|---|---|---|
| Input channels | 100 | 100 |
| Model dimension | 512 | 1536 |
| Intermediate dimension | 1536 | 4608 |
| Number of layers | 8 | 8 |
| Phase quantization levels | 128 | 128 |

| MPD | | |
|---|---|---|
| Periods $P_k$ | | [2, 3, 5, 7, 11] |

| MRD / cMRD | | |
|---|---|---|
| FFT sizes | | [512, 1024, 2048] |
| Hop sizes | | [128, 256, 512] |
| Window sizes | | [512, 1024, 2048] |
| Bands ratio | | [0, 0.1, 0.25, 0.5, 0.75, 1.0] |

| Training | Base | Large |
|---|---|---|
| Batch size | 16 | 32 |
| Steps | 1M | 1M |
| Segment size | 16,384 | 16,384 |
| Initial learning rate | 2e-4 | 2e-4 |
| Scheduler | cosine | cosine |
| Optimizer | AdamW | AdamW |
| $\beta_1, \beta_2$ | (0.8, 0.9) | (0.8, 0.9) |
| $\lambda_{\text{Mel}}$ | 45 | 45 |
| $\lambda_{\text{MPD}}$ | 1.0 | 1.0 |
| $\lambda_{\text{cMRD}}$ | 0.1 | 0.1 |

| Hardware | | |
|---|---|---|
| GPU | | $1\times$ NVIDIA A6000 |
| CPU | | Intel Xeon Gold 6148 @ 2.40 GHz |

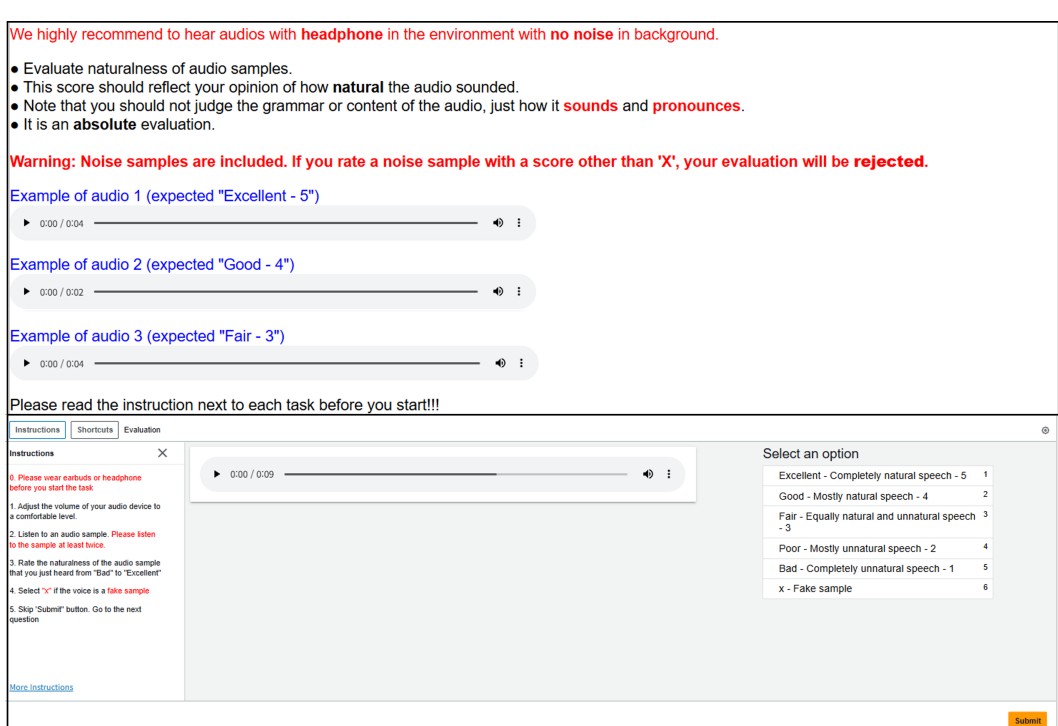

Figure 6: MOS evaluation interface.

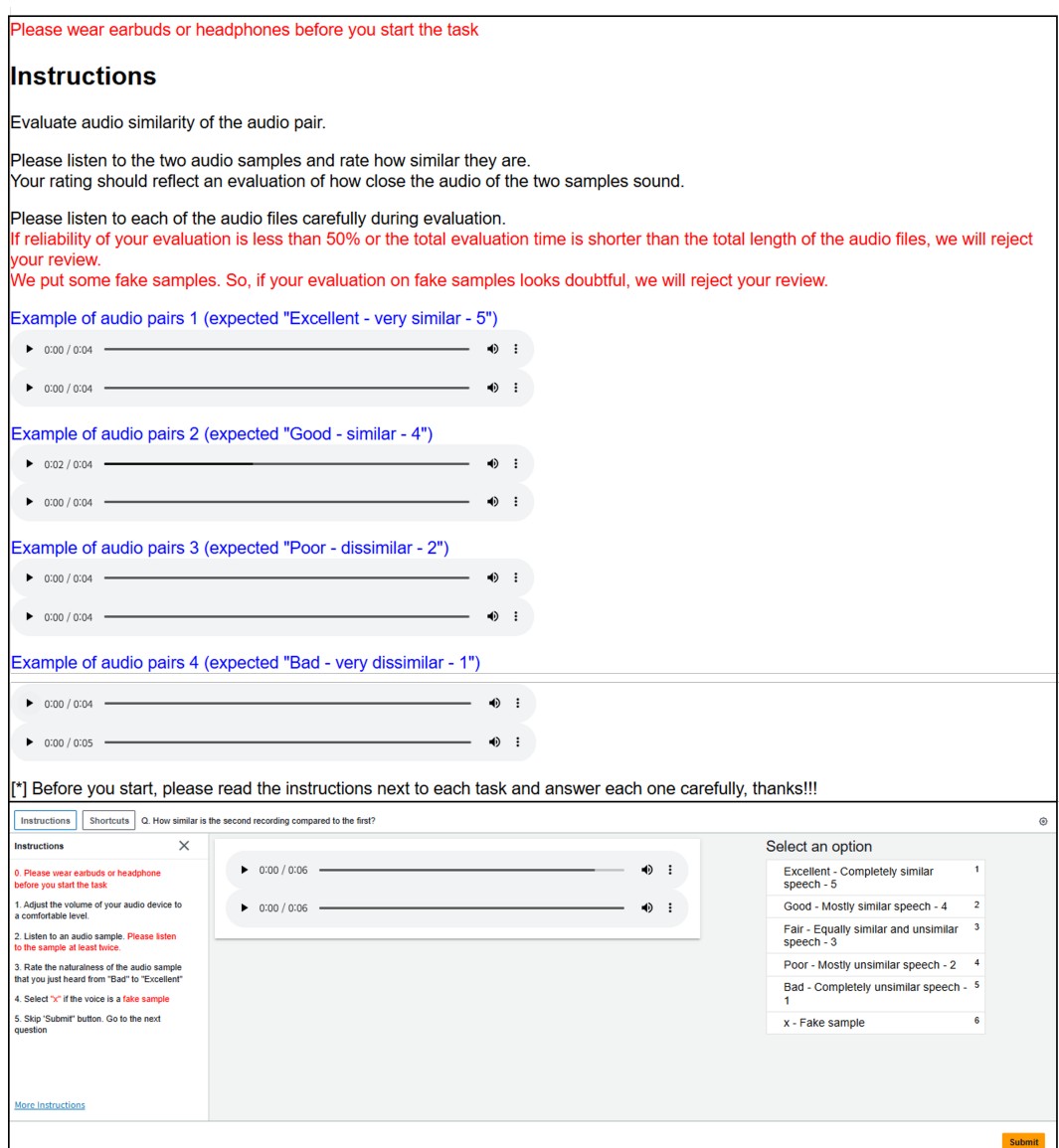

Figure 7: SMOS evaluation interface.

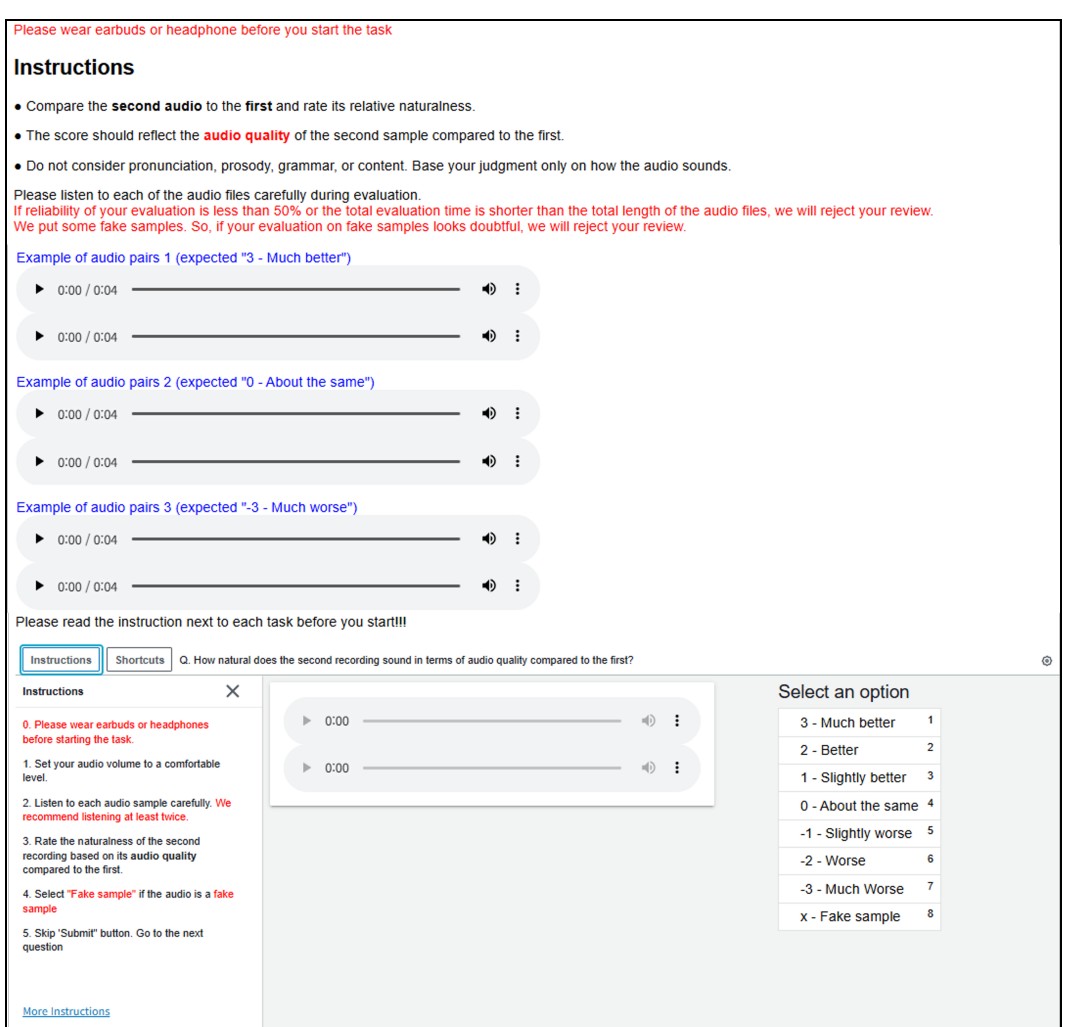

Figure 8: CMOS evaluation interface.

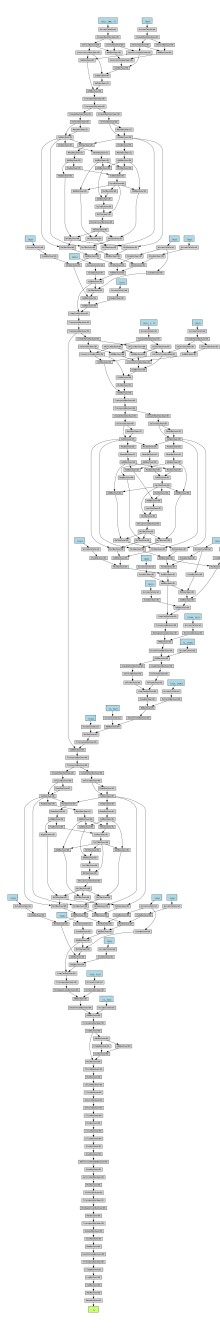

Figure 9: Backward computation graph of the generator using the native PyTorch complex imple-
mentation.

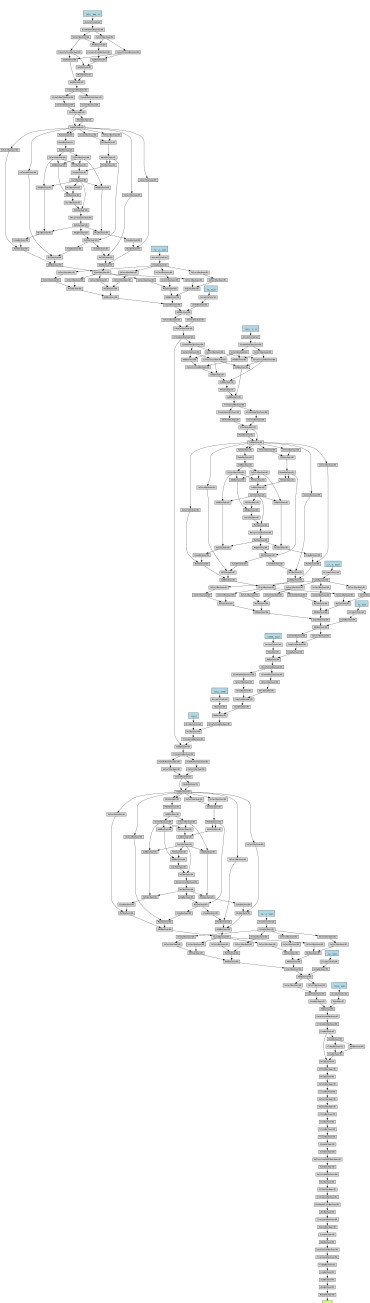

Figure 10: Backward computation graph of the generator using Gauss' multiplication trick.

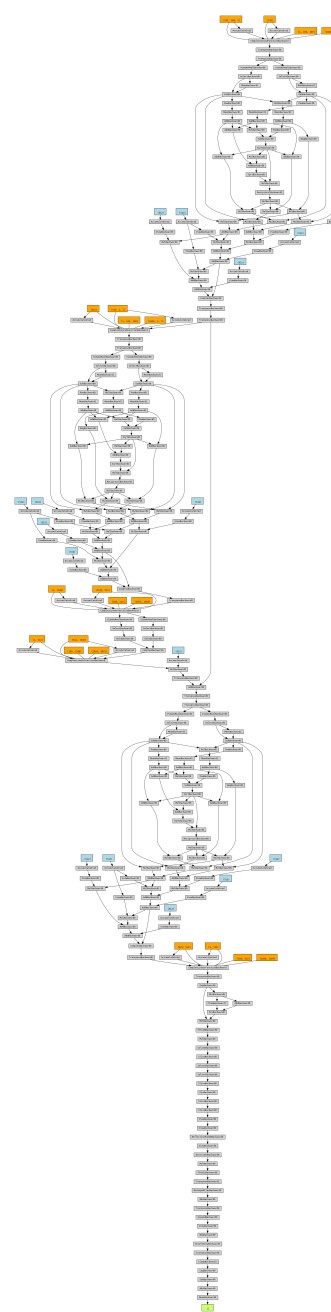

Figure 11: Backward computation graph of the generator using the block-matrix operation.

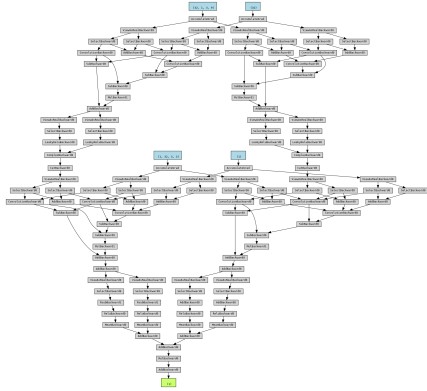

Figure 12: Backward computation graph of the cMRD using the native PyTorch complex implementation.

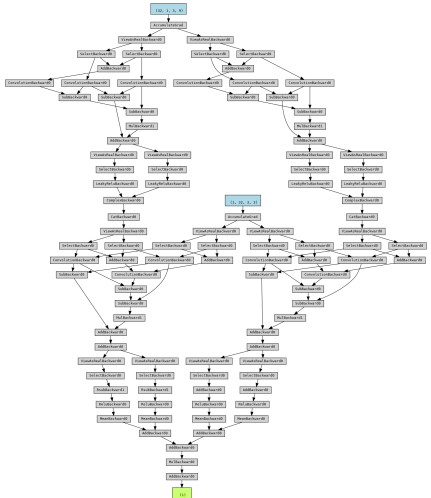

Figure 13: Backward computation graph of the cMRD using Gauss' multiplication trick.

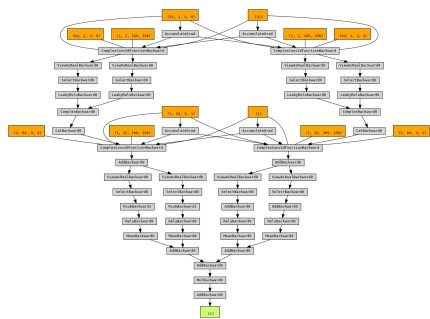

Figure 14: Backward computation graph of the cMRD using the block-matrix operation.

