# OpenReview forum: "Toward Complex-Valued Neural Networks for Waveform Generation"
_ICLR.cc/2026/Conference — ICLR 2026 Poster_

### Official Review · Reviewer_GeoB · 2025-10-27

**Soundness:** 3
**Presentation:** 4
**Contribution:** 3
**Rating:** 4
**Confidence:** 3

**Summary:**

This paper introduces ComVo, a complex-valued neural vocoder designed to generate high-fidelity audio within the iSTFT-based vocoder framework. ComVo leverages complex-valued neural networks (CVNNs) for both the generator and discriminators, enabling better modeling of signal characteristics and thereby enhancing audio reconstruction quality. In addition, two practical techniques are proposed: (1) Phase quantization - stabilizes training by mitigating abrupt phase drift, and (2) Block-matrix computation - improves computational efficiency by performing block-wise operations on real and imaginary components of complex-valued tensors. Experimental results show that ComVo outperforms real-valued vocoders with comparable parameter counts across multiple evaluation metrics (e.g., UTMOS, PESQ, MR-STFT). Further ablation studies demonstrate the effect of applying the phase quantization and block-matrix computation.

**Strengths:**

- The use of CVNNs demonstrates clear performance gains over real-valued architectures.
- The paper presents various ablation studies, providing insight into the role of each proposed component in the overall system. The architectural design is well-motivated and supported by empirical evidence.
- ComVo scales effectively, showing competitive performance across both lightweight and large model configurations.
- The advantage of ComVo is maintained even when integrated into a TTS pipeline, highlighting its versatility and robustness.

**Weaknesses:**

- For me, the proposed block-matrix computation seems more like an implementation-level optimization rather than a novel research contribution.
- The ablation results for phase quantization appear relatively weak, offering limited evidence to justify its effectiveness.
- Although Table 8 indicates that ComVo and the baselines have similar parameter counts, ComVo uses a different parameter datatype (e.g., ComVo’s parameters are stored in complex64 format, which effectively corresponds to two float32 tensors for real and imaginary parts). Therefore, for a fair comparison of model capacity and memory usage, the baseline parameter counts should be doubled.

Overall, the ComVo model itself is competitive and valuable, and its performance improvements are noteworthy. However, given that ‘phase quantization’ and ‘block-matrix computation’ are addressed as major contributions in the main text, I think their novelty and impact should be more clarified and empirically supported to substantiate their significance as research contributions.

**Questions:**

- The phase quantization layer is applied after the first complex Conv1D in the generator. If the first Conv1D was not complex-valued, the phase quantization layer would not be required. What would happen if this sequence (complex Conv1D followed by phase quantization) was replaced with a single real-valued Conv1D layer? Such a comparison would help clarify the necessity and contribution of the phase quantization layer.
- Why is the phase quantization layer not commonly employed in other complex-valued CNNs?
- Could the authors provide a theoretical justification for why the phase quantization layer is beneficial or necessary for improved performance?
- Why the inference latency of ComVo is noticeably slower than Vocos, despite adopting a similar architectural foundation?
- A comparison of models with equivalent memory usage would be informative, since ComVo uses complex-valued tensors for its parameters, which actually consist of two float tensors. For example, it would be interesting to see whether a ComVo model with half the number of parameters still outperforms the real-valued baselines.

---

> ### Author Response · Authors · 2025-11-21
> **Response to Reviewer Comments**
>
> We appreciate the reviewer’s careful evaluation of our work and the helpful comments provided.
>
> ### Comment 1
>
> > The proposed block-matrix computation seems more like an implementation-level optimization rather than a novel research contribution.
>
> Thank you for your interest in the block-matrix computation component. As research on CVNNs is still at an early stage, there remain many methodological challenges that need to be explored. Just as the early phases of deep learning research involved substantial work on improving training stability and computational efficiency, we believe that similar efforts are essential for advancing complex-valued neural networks as well.
>
> Existing work has already explored ways to reduce the cost of complex-valued operations. One example is Gauss’ multiplication trick, which is used in libraries such as *complextorch* to lower the arithmetic cost of complex multiplication. To understand how our approach relates to these existing strategies, we conducted an additional comparison of several execution methods for complex-valued layers.
>
> Specifically, we examined native PyTorch complex operations, a Gauss-based method, and our block-matrix formulation under identical models and inputs. The Gauss-based method showed only minor differences relative to the native method, while the block-matrix formulation consistently reduced overall training cost by offering a lighter backward computation profile for both the generator and discriminator. Since backward computation dominates CVNN training time, this leads to a noticeable practical advantage.
>
> We view this as a modest but useful step toward improving the computational feasibility of CVNNs, and we hope it encourages further exploration of efficiency-oriented techniques in future work.
>
> | Implementation     | Gen Forward (s) | Gen Backward (s) | Disc Forward (s) | Disc Backward (s) |
> |--------------------|------------------|-------------------|-------------------|--------------------|
> | Native PyTorch     | 0.005288         | 0.234591          | 0.079073          | 0.190067           |
> | Gaussian trick     | 0.005160         | 0.231545          | 0.074103          | 0.184894           |
> | Block-matrix       | 0.005786         | 0.181283          | 0.050389          | 0.139159           |

---

> ### Author Response · Authors · 2025-11-21
> **Response to Reviewer Comments**
>
> ### Comment 2
>
> > Could the authors provide a theoretical justification for why the phase quantization layer is beneficial or necessary for improved performance?
> >
>
> > The phase quantization layer is applied after the first complex Conv1D in the generator. If the first Conv1D was not complex-valued, the phase quantization layer would not be required. What would happen if this sequence (complex Conv1D followed by phase quantization) was replaced with a single real-valued Conv1D layer? Such a comparison would help clarify the necessity and contribution of the phase quantization layer.
> >
>
> > Why is the phase quantization layer not commonly employed in other complex-valued CNNs?
> >
>
> **(1) Theoretical motivation for phase quantization**
>
> Thank you for the question about why phase quantization (PQ) is used and how we determined its placement. Since our generator receives real-valued inputs, the imaginary component of the complex representation must be created from scratch. At this stage, the model can produce arbitrary phase because there is no physical or data-driven constraint guiding how the initial phase should form, which can lead to unnecessary variability and unstable gradients. Prior work has also noted that fully unconstrained phase is not always desirable [1, 2]. Motivated by this, we add a PQ step at the point where complex features first emerge to impose a small structural constraint and regularize the initially random phase formation, while allowing the subsequent layers to operate without explicit restrictions.
>
> [1] W. M. Eric et al., “Phase modeling and quantization for low-rate harmonic/noise coding,” *European Signal Processing Conference*, 2002.
>
> [2] D.-S. Kim “Perceptual phase quantization of speech,” *IEEE transactions on speech and audio processing*, 2003.
>
> **(2) Ablation on replacing the first complex Conv1D + PQ with a real Conv1D**
>
> We agree that the initial complex layer could influence how the effect of PQ is interpreted. Following the reviewer’s suggestion, we trained a variant where the first complex Conv1D was replaced with a real Conv1D that outputs two real channels, which are then treated as the real and imaginary parts to construct a complex feature. The rest of the generator remains unchanged. This variant trains properly and performs similarly to the version without PQ, while the original configuration yields a different result. This comparison helps isolate the role of the PQ operation and shows that its effect is tied to its interaction with the first complex layer. We have incorporated this analysis into the revised manuscript.
> We also tested placing the same quantization step at deeper layers.
> The ConvNeXt block follows the sequence
>
> [Depthwise Conv] – 1 –[Pointwise Conv] – 2 – [GELU] – 3 – [Pointwise Conv] – 4
>
> and we inserted PQ at positions 1, 2, 3, and 4 within this structure. These placements produced inconsistent outcomes and sometimes degraded performance, suggesting that applying PQ beyond the initial complex layer does not consistently provide benefits.
>
> | Model              | UTMOS  | MR-STFT | PESQ   | Periodicity | V/UV F1 |
> |--------------------|--------|---------|--------|-------------|---------|
> | 0                  | 3.6646 | 0.8435  | 3.7756 | 0.0915      | 0.9625  |
> | Real Conv          | 3.6337 | 0.8610  | 3.7774 | 0.0980      | 0.9574  |
> | Complex Conv + PQ  | 3.6901 | 0.8439  | 3.8239 | 0.0903      | 0.9609  |
>
> | Model | UTMOS  | MR-STFT | PESQ   | Periodicity | V/UV F1 |
> |-------|--------|---------|--------|-------------|---------|
> | ComVo | 3.6901 | 0.8439  | 3.8239 | 0.0903      | 0.9609  |
> | 1     | 3.6110 | 0.8684  | 3.7347 | 0.0954      | 0.9583  |
> | 2     | 3.6171 | 0.8693  | 3.7103 | 0.0914      | 0.9610  |
> | 3     | 3.6174 | 0.8697  | 3.7372 | 0.0959      | 0.9575  |
> | 4     | 3.6643 | 0.8622  | 3.7780 | 0.0873      | 0.9649  |
>
> **(3) Why PQ is not commonly used in other CVNNs**
>
> PQ has not appeared in prior CVNN architectures simply because it has not been explored in that setting. Our work introduces PQ as a new way to impose structure on phase behavior when forming complex-valued features, and we show that it leads to improvements in our model.
>
> Although earlier studies outside the CVNN literature have explored phase quantization or discretization [1–3], using this idea within a complex-valued neural network has not, to our knowledge, been investigated before. We believe the concept may be useful more broadly in CVNN tasks that involve learning complex-valued representations from data.
>
> [1] W. M. Eric et al., “Phase modeling and quantization for low-rate harmonic/noise coding,” *European Signal Processing Conference*, 2002.
>
> [2] D.-S. Kim “Perceptual phase quantization of speech,” *IEEE transactions on speech and audio processing*, 2003.
>
> [3] N. Takahashi et al., “PhaseNet: Discretized Phase Modeling with Deep Neural Networks for Audio Source Separation.,” *INTERSPEECH*, 2018.

---

> ### Author Response · Authors · 2025-11-21
> **Response to Reviewer Comments**
>
> ### Comment 4
>
> > Why the inference latency of ComVo is noticeably slower than Vocos, despite adopting a similar architectural foundation
>
>
> Thank you for the question regarding inference latency.  Although ComVo and Vocos share a similar backbone, their runtime differs because complex-valued convolutions generally have higher computational cost than real-valued ones. Current deep-learning frameworks also provide limited optimization for complex operations, so the difference becomes more noticeable in inference settings. Overall, the latency gap reflects the present level of framework support for complex-valued computation rather than a limitation of the model itself.
>
> ### Comment 5
>
> > A comparison of models with equivalent memory usage would be informative, since ComVo uses complex-valued tensors for its parameters, which actually consist of two float tensors. For example, it would be interesting to see whether a ComVo model with half the number of parameters still outperforms the real-valued baselines.
>
>
> Thank you for raising the point about comparing models under matched memory usage. We agree that a fair comparison should account for the effective memory usage of complex-valued parameters. For this reason, we already included a memory-matched experiment in the submitted paper. As shown in Table 9, we constructed a widened real-valued baseline (GRDR 2×) whose parameter count and memory footprint closely match those of the complex-valued generator.
> Even under this matched-memory setting, the complex-valued model still outperformed the enlarged real-valued baseline across all evaluation metrics. This comparison is already part of the paper to show that the performance gap is not attributable to parameter capacity, but to the modeling approach itself.

---

> > ### Comment · Reviewer_GeoB · 2025-11-27
> >
> > Thank you for detailed explanations. I recommend to include the additional experiments in the appendix. I will raise the score.

---

> > > ### Author Response · Authors · 2025-11-28
> > >
> > > Thank you for acknowledging our efforts and for providing such insightful comments.
> > > Your feedback has directly helped us improve the clarity and completeness of the paper.
> > > We will incorporate the additional experiments into the appendix and update the manuscript accordingly.
> > > We truly appreciate your careful review.

---

### Official Review · Reviewer_gyMx · 2025-10-30

**Soundness:** 3
**Presentation:** 3
**Contribution:** 2
**Rating:** 6
**Confidence:** 5

**Summary:**

This paper introduces ComVo, a complex-valued neural vocoder that performs waveform generation entirely in the complex domain using a GAN-based architecture. The authors argue that existing iSTFT-based vocoders rely on real-valued networks that process real and imaginary parts independently, limiting their ability to capture the inherent structure of complex spectrograms. The proposed model employs complex-valued neural networks (CVNNs) to jointly model the real and imaginary components of spectrograms. The paper also introduces phase quantization as a regularization method and a block-matrix computation scheme to improve training efficiency. The experimental results show that ComVo outperforms existing real-valued vocoders in terms of synthesis quality and training time.

**Strengths:**

Originality: The introduction of complex-valued neural networks (CVNNs) for waveform generation is an interesting and novel approach that is not widely explored in the context of vocoders. The proposed method shows potential in capturing the structure of complex spectrograms by treating them as unified complex entities.

Quality: The paper is well-written, and the experimental setup is clearly described. The proposed method shows promising results in terms of both objective and subjective evaluations.

Clarity: The paper is presented in a clear and structured manner. The explanations of the method, including the details of the generator, discriminator, phase quantization, and block-matrix computation scheme, are well-explained.

**Weaknesses:**

A major weakness of this paper is that it compares the proposed method only to real-valued vocoders and iSTFT-based methods. The paper does not include a comparison with vocoders that predict both amplitude and phase spectrograms (such as APNet and FreeV). These methods already integrate both real and imaginary parts in their amplitude and phase spectrogram predictions, which might address the issue the authors claim with real-valued networks. Without this comparison, it is difficult to conclusively prove that ComVo offers a significant advantage over existing methods. The authors should include such comparisons to strengthen the argument for their method's effectiveness.

**Questions:**

See the above Weaknesses.

---

> ### Author Response · Authors · 2025-11-21
> **Response to Reviewer Comments**
>
> We sincerely appreciate the reviewer’s thoughtful feedback and valuable suggestions.
> ### Comment 1
>
> > A major weakness of this paper is that it compares the proposed method only to real-valued vocoders and iSTFT-based methods. The paper does not include a comparison with vocoders that predict both amplitude and phase spectrograms (such as APNet and FreeV).
>
> We appreciate your suggestion to include amplitude–phase prediction approaches in the comparison. To address this, we conducted additional experiments including representative models such as APNet, APNet2, and FreeV. We trained APNet, APNet2, and FreeV with their official implementations using the same data configuration as ours.
>
> These methods estimate amplitude and phase explicitly using real-valued architectures, where the two components are treated as separate regression targets. In contrast, our work investigates an alternative parameterization in which the generator models the complex spectrogram directly in the complex domain.
>
> This difference in representation led to improved performance in our experiments, indicating that the complex-domain formulation is effective for this task. We appreciate your suggestion, and the updated version of the manuscript now reflects the additional results and clarification.
>
> | Model  | UTMOS  | MR-STFT | PESQ   | Periodicity | V/UV F1 |
> |--------|--------|---------|--------|-------------|---------|
> | APNet  | 2.4015 | 1.3375  | 2.8457 | 0.1582      | 0.9185  |
> | APNet2 | 2.7379 | 1.1582  | 2.7748 | 0.1448      | 0.9243  |
> | FreeV  | 2.6971 | 1.1782  | 2.7960 | 0.1581      | 0.9105  |
> | ComVo  | 3.6901 | 0.8439  | 3.8239 | 0.0903      | 0.9609  |

---

### Official Review · Reviewer_eE9R · 2025-10-31

**Soundness:** 2
**Presentation:** 2
**Contribution:** 2
**Rating:** 4
**Confidence:** 4

**Summary:**

This paper introduces ComVo, a neural vocoder for waveform generation that operates within the complex domain. The core idea is to leverage complex-valued neural networks (CVNNs) for both the generator and a multi-resolution discriminator (cMRD), arguing that this allows the model to better capture the intrinsic structure of complex spectrograms compared to conventional real-valued networks that process real and imaginary components independently. The authors also introduce two technical refinements: a phase quantization layer to act as a regularizer, and a block-matrix computation scheme to improve training efficiency. The paper presents a series of experiments and ablation studies showing that ComVo achieves competitive or superior performance on objective and subjective metrics against several strong real-valued vocoder baselines.

**Strengths:**

The exploration of complex-valued networks for generative audio tasks is a compelling research direction, and the authors present a well-executed implementation. The paper is technically solid; the proposed phase quantization is an interesting inductive bias for stabilizing phase prediction, and the block-matrix formulation for accelerating training is a valuable engineering contribution that demonstrably reduces training time by 25%. The experimental evaluation is thorough, with comparisons against strong, widely-used baselines like HiFi-GAN and Vocos on standard datasets. The ablation studies in Table 4 are particularly useful for dissecting the contributions of the complex-valued generator and discriminator.

**Weaknesses:**

Despite the positive results, I have fundamental reservations about the paper's central motivation. The primary claim is that CVNNs are superior because they "capture the intrinsic dependencies between the real and imaginary components." However, this central hypothesis is asserted rather than rigorously validated. The performance gains, while present, do not in themselves prove that this specific mechanism is the cause. My main conceptual issue is that for a spectrogram to be perfectly invertible back to a real-valued signal, it must satisfy time-frequency consistency. This means the space of valid spectrograms for real audio is a highly structured subspace within the broader domain of all possible complex spectrograms. By moving all computations into an unconstrained complex domain, the model may actually face the additional burden of learning to stay within this physically valid subspace, which might not be an advantage. The paper does not address this potential conflict.
Furthermore, the overall architecture still relies on a standard Multi-Period Discriminator (MPD) operating on the real-valued waveform. The MPD is known to be a very powerful component in modern vocoders. Its presence makes it difficult to ascertain whether the observed quality improvements truly stem from the benefits of complex-domain feedback via the cMRD, or if the MPD is still doing the majority of the perceptual heavy lifting. Finally, from a structural standpoint, the detailed background in Section 2.1 on the fundamentals of CVNNs feels more appropriate for an appendix, as it disrupts the main narrative of the paper.

**Questions:**

I hope the authors can address the following points to strengthen their claims and clarify the contributions of their work:
1. The core premise of the paper needs a stronger defense. Given that the ultimate target is a real-valued signal, which implies its spectrogram must satisfy time-frequency consistency, why is it advantageous for the hidden layers to operate under general complex arithmetic rather than in a way that respects this physical constraint from the outset? Could the unconstrained complex modeling actually be a less efficient path to the desired solution space?
2. The central claim is that the model better "captures intrinsic dependencies" between real and imaginary parts. Beyond the final performance metrics, is there any more direct analysis you can provide to support this? For instance, have you analyzed the learned phase-magnitude relationships or the structure of the internal representations to show they are more coherent than in real-valued models?
3. The ablation study in Table 4 is insightful, but the powerful real-valued MPD remains a constant across all configurations. How can we be confident that the gains attributed to the complex-valued cMRD are not simply an artifact of having two strong discriminators, with the MPD still being the primary driver of quality? Have you considered an ablation where only the cMRD is used, to truly isolate its effectiveness?
4. Other iSTFT-based models like Vocos and iSTFTNet also operate on complex spectrograms, but their motivation seems more direct—it's a computationally efficient and direct target for the network. Your paper claims a more fundamental modeling advantage. Could you elaborate on why your motivation leads to a better vocoder than one motivated purely by computational efficiency?

---

> ### Author Response · Authors · 2025-11-21
> **Response to Reviewer Comments**
>
> We thank the reviewer for the insightful comments and constructive suggestions.
>
> ### Comment 1
>
> > Given that the ultimate target is a real-valued signal, which implies its spectrogram must satisfy time-frequency consistency, why is it advantageous for the hidden layers to operate under general complex arithmetic rather than in a way that respects this physical constraint from the outset? Could the unconstrained complex modeling actually be a less efficient path to the desired solution space?
>
> We appreciate your clarification regarding this point. As you pointed out, the final output is a real-valued signal, but the model is trained to predict a complex spectrogram, and the consistency requirements of the STFT apply only to this final prediction. As in prior iSTFT-based vocoders, the hidden layers do not operate on STFT-consistent representations and are not expected to do so, since enforcing such constraints inside the network is neither feasible nor part of standard practice in spectrogram-based generation models.
>
> Within the hidden layers, complex arithmetic does not introduce an unconstrained search space. Instead, complex-valued layers provide a structured way of keeping the real and imaginary parts coupled, which is more aligned with the form of the target spectrogram than representing them as two independent real channels. Prior work on complex-valued neural networks has similarly reported that learning directly in the complex domain can lead to more coherent internal transformations and favorable optimization behavior when the predicted output itself is complex [1–3].
>
> In our setting, using complex layers allows the model to learn internal representations that naturally reflect the structure of a complex spectrogram, without splitting the real and imaginary parts into separate channels. The final consistency constraint is handled entirely by the iSTFT, while the hidden layers benefit from a representation that matches the nature of the prediction target.
>
> [1] A. M. Sarrof, "Complex Neural Networks for Audio," *PhD Thesis. Dartmouth College*, 2018.
>
> [2] J. A. Barrachina et al., "Complex-Valued Vs. Real-Valued Neural Networks for Classification Perspectives: An Example on Non-Circular Data," *ICASSP*, 2021.
>
> [3] K. S. Mayer et al., "Adaptive Learning Rate Methods for Complex-Valued Neural Networks," IEEE *Transactions on Neural Networks and Learning Systems*, 2025.
>
> ### Comment 2
>
> > The central claim is that the model better "captures intrinsic dependencies" between real and imaginary parts. Beyond the final performance metrics, is there any more direct analysis you can provide to support this? For instance, have you analyzed the learned phase-magnitude relationships or the structure of the internal representations to show they are more coherent than in real-valued models?
>
> We appreciate your request for a more direct analysis of whether the model captures dependencies between the real and imaginary components. To isolate this question from the full vocoder setting, we conducted a controlled experiment inspired by prior work showing that complex-valued networks can more effectively model structured complex quantities [1].
>
> In this experiment, we implemented lightweight MLP-based generators and discriminators in both CVNN and RVNN form. To ensure a fair comparison of expressive capacity, the RVNN variants were configured with twice as many parameters as the corresponding CVNN models. Each model was trained across ten random seeds on a synthetic complex-valued target distribution, and evaluated using Jensen–Shannon divergence (JSD) applied separately to magnitude and phase. On average, the CVNN achieved lower JSD for both measures, showing that it captured the magnitude–phase structure more accurately.
>
> This experiment suggests that CVNNs represent complex-domain structure in a more organized way and show closer alignment with magnitude–phase relationships under this comparison. We appreciate the reviewer’s comment, which motivated us to include an additional experiment that further supports our claim. The detailed description of this experiment, along with the corresponding visualizations, has been incorporated into the updated manuscript.
>
> [1] J. A. Barrachina et al., "Complex-Valued Vs. Real-Valued Neural Networks for Classification Perspectives: An Example on Non-Circular Data," ICASSP, 2021.
>
> | Model | JSD (magnitude)         | JSD (phase)             |
> |-------|--------------------------|--------------------------|
> | RVNN  | 0.018350 ± 0.014447      | 0.021110 ± 0.036340      |
> | CVNN  | 0.006548 ± 0.002867      | 0.003911 ± 0.002181      |

---

> ### Author Response · Authors · 2025-11-21
> **Response to Reviewer Comments 2**
>
> ### Comment 3
>
> > The ablation study in Table 4 is insightful, but the powerful real-valued MPD remains a constant across all configurations. How can we be confident that the gains attributed to the complex-valued cMRD are not simply an artifact of having two strong discriminators, with the MPD still being the primary driver of quality? Have you considered an ablation where only the cMRD is used, to truly isolate its effectiveness?
>
> We understand the your concern that the improvement might look like a side effect of using two discriminators, with the MPD still being the main component. To clarify this, we conducted an expanded set of ablations including MPD only, MRD only, cMRD only, MPD + MRD, and MPD + cMRD, all trained under identical conditions.
>
> As observed in earlier studies, removing the MPD results in lower time-domain quality. This is why many existing systems pair the MPD with an STFT-domain discriminator, and our baseline architecture follows this common design.
>
> We recognize that the MPD could obscure the effect of the cMRD, so we conducted additional ablations to examine its role directly. In the MPD-free setting, the cMRD alone outperforms the real-valued MRD, showing that its contribution is independent of the MPD. These results have been incorporated into the revised manuscript.
>
> | Model        | UTMOS  | MR-STFT | PESQ   | Periodicity | V/UV F1 |
> |--------------|--------|---------|--------|-------------|---------|
> | MPD only     | 3.6357 | 0.8522  | 3.7670 | 0.0942      | 0.9613  |
> | MRD only     | 2.8338 | 0.8442  | 3.9868 | 0.0870      | 0.9610  |
> | cMRD only    | 2.9285 | 0.8398  | 4.0149 | 0.0859      | 0.9635  |
> | MPD + MRD    | 3.6452 | 0.8597  | 3.7375 | 0.0978      | 0.9567  |
> | MPD + cMRD   | 3.6901 | 0.8439  | 3.8239 | 0.0903      | 0.9609  |
>
> ### Comment 4
>
> > Other iSTFT-based models like Vocos and iSTFTNet also operate on complex spectrograms, but their motivation seems more direct—it's a computationally efficient and direct target for the network. Your paper claims a more fundamental modeling advantage. Could you elaborate on why your motivation leads to a better vocoder than one motivated purely by computational efficiency?
>
> Thank you for pointing out this aspect of iSTFT-based vocoders. Prior iSTFT-based vocoders such as Vocos and iSTFTNet adopt the complex spectrogram primarily for computational efficiency. By operating at the frame level, these systems avoid heavy waveform upsampling and achieve very fast inference, making efficiency the central motivation behind their design.
>
> In iSTFT-based methods in general, the model outputs a complex spectrogram that is converted back to a waveform through the inverse STFT. This is the standard output format used in iSTFT-based vocoders.
>
> Our work focuses on a different aspect. Like prior iSTFT-based vocoders, we retain the efficiency benefits of predicting at the frame level without waveform upsampling. At the same time, generating a complex spectrogram raises the issue of how this complex representation should be modeled within the network. Real-valued architectures are widely used and strong baselines, but they represent the complex spectrogram indirectly by separating it into two real-valued channels.
>
> We explore an alternative parameterization that models the spectrogram directly in the complex domain using complex-valued layers. This provides a representation aligned with the form of the output and, in our experiments, leads to more effective modeling behavior under comparable conditions.
>
> In summary, earlier iSTFT-based vocoders adopt the complex spectrogram mainly for efficiency. Our contribution instead concerns how the complex spectrogram itself is modeled, and we find that a complex-valued parameterization offers a natural and empirically beneficial choice for this purpose.

---

### Author Response · Authors · 2025-12-04
**Review Summary and Revisions**

We thank the Area Chair and reviewers for their feedback. The reviews highlighted the **originality of using complex-valued neural networks for vocoding**, the **architecture design**, and the **evaluation** with strong baselines and ablations.
### **Summary of main concerns**
(1) **Justification and evidence for complex-domain modeling and the complex discriminator**

(2) **The need for additional experimental comparisons**, especially with amplitude–phase vocoders

(3) **Design and impact of the phase quantization (PQ) layer and block-matrix computation**, including efficiency and memory usage

### **Summary of our response**
(1) **Complex-domain modeling & complex discriminator**: Added a CVNN vs. RVNN comparison on a controlled synthetic complex distribution. Extended discriminator ablations to isolate the contribution of the complex multi-resolution discriminator (cMRD), both with and without MPD.

(2) **Additional experimental comparisons**: Included additional amplitude–phase vocoder baselines: APNet, APNet2, and FreeV.

(3) **Phase quantization & block-matrix scheme**: Conducted PQ ablations varying (i) real vs. complex first layer and (ii) PQ placement in the network. Analyzed the block-matrix computation scheme, reporting latency, memory footprint, and comparisons against memory-matched baselines.

Reviewer GeoB acknowledged these additions and raised their score; the discussion window closed before Reviewers eE9R and gyMx could respond.

---
### **Reviewer eE9R**

| Concern | Response | Location in Paper |
| --- | --- | --- |
| **Justification for complex-valued modeling under STFT consistency** | We clarified that, as in iSTFT-based vocoders, the consistency requirement applies only to the final complex spectrogram passed to iSTFT; intermediate features are not constrained to be STFT-consistent. In this setting, complex-valued layers process real and imaginary parts jointly as one complex quantity rather than as separate real channels. We also shortened the CVNN background and moved detailed material to an appendix. | **Sec. 2.1; Appendix A** |
| **Direct evidence for complex-domain modeling** | We added a synthetic complex-distribution experiment comparing lightweight CVNN and RVNN models, with RVNNs given 2× parameters. Under the same setup, CVNNs obtain lower JSD on magnitude and phase, supporting better complex-domain modeling than RVNNs. | **Sec. 3; Appendix B** |
| **Contribution of cMRD beyond MPD** | We extended the discriminator ablation to MPD only, MRD only, cMRD only, MPD+MRD, and MPD+cMRD. cMRD outperforms MRD both with and without MPD, showing that the complex discriminator contributes beyond simply adding a second discriminator alongside MPD. | **Sec. 5.3** |

---
### **Reviewer gyMx**

| Concern | Response | Location in Paper |
| --- | --- | --- |
| **Comparison with amplitude–phase vocoders** | We added APNet, APNet2, and FreeV as baselines, trained with their official implementations under the same data configuration. ComVo consistently outperforms these amplitude–phase vocoders across all metrics, showing that our advantage extends to methods that explicitly predict amplitude and phase. | **Sec. 2.2; Appendix J** |

---
### **Reviewer GeoB**

| Concern | Response | Location in Paper |
| --- | --- | --- |
| **Analysis of phase quantization and its placement** | We clarified PQ as a light structural constraint on the first complex layer, where the imaginary component is created from real-valued inputs. We added ablations that (i) replace the first complex Conv1D+PQ with a real Conv1D outputting two channels interpreted as (Re, Im), and (ii) insert PQ at different positions inside the ConvNeXt block. The real-Conv variant behaves similarly to the model without PQ, and later PQ placements show no consistent improvement over this configuration, indicating that PQ is most effective where complex features first emerge. | **Sec. 5.4; Appendix I** |
| **Efficiency gains from block-matrix computation** | We compared native PyTorch complex ops, a Gauss-based implementation, and our block-matrix scheme under identical models and inputs. The block-matrix scheme keeps forward cost similar but substantially reduces backward computation time for both generator and discriminator, and we provided a short derivation and numerical checks showing equivalence to standard complex operations. | **Sec. 5.5; Appendices D–F** |
| **Computational efficiency and memory-matched comparison** | We clarified that the submission already includes a memory-matched real-valued baseline (GRDR 2×) with parameter count and memory footprint comparable to the complex generator. GRDR 2× improves over the base GRDR but still underperforms the complex generator (GCDR), supporting that the gains come from the modeling choice rather than capacity alone. | **Sec. 5.7; Appendix H** |

We hope this summary of reviewer feedback, our responses, and the corresponding revision locations is helpful for your assessment.

---

### Meta-Review · Area_Chair_amtX · 2026-01-08

**Summary:**

The reviewers generally recognize that the paper presents a novel and technically solid exploration of complex-valued neural networks (CVNNs) for vocoder design, with all three highlighting competitive performance gains over strong real-valued baselines and clear engineering contributions(phase quantization and block-matrix computation). Key points across the reviews include the need for more direct evidence supporting the complex-domain modeling advantage (Reviewer 1), comprehensive comparisons with amplitude–phase predictive vocoders (Reviewer 2), and clarification of the practical and conceptual impact of proposed components like phase quantization and block-matrix computation (Reviewer 3). The rebuttal effectively addresses these concerns with additional ablations, controlled experiments, and comparisons to matched-memory baselines, demonstrating that the performance improvements are due to the complex-domain modeling rather than confounding factors. Reviewer 3 increased rating eventually. Some reviewers note that certain contributions (e.g., block-matrix computation) are more engineering-oriented and the novelty of phase quantization could be further highlighted, the overall methodological rigor and empirical validation are convincing.

Given that the final reviewer ratings cluster around borderline but positive (4, 6, 4) and all major concerns have been adequately addressed in the rebuttal, I recommend a weak accept.

Several sections of this paper has blue highlighted texts, suggesting an incomplete paper revision (?) Is the paper submitted in rush? And no review pointed this out.

**Reviewer Concerns:**

Reviewer 1 concerns are overall addressed, and some conceptual skepticism remains about whether unconstrained complex modeling is fundamentally better.

Reviewer 2 concerns are also addressed by added experiments comparing ComVo to amplitude–phase vocoders (APNet, FreeV).

Reviewer 3 concerns are addressed, with raised rating. Some perception of novelty for PQ and block-matrix is still subjective; minor latency concerns remain.

**Reviewer Scores:**

Reviewer 1 would retain original rating or increase to 5 slightly, as rebuttal addressed main conceptual concerns and provided ablations.

Reviewer 2 would retain original rating, as the original rating is high.

Reviewer 3 has already increased rating after rebuttal.

---

### Decision · Program_Chairs · 2026-01-26

Accept (Poster)